# Explaining the Complex Task Reasoning of Large Language Models with Template-Content Structure

## Abstract

The continuous evolution of pre-trained large language models with ever-growing parameters and corpus sizes has augmented their capacity to solve complex tasks. This ability, which obviates the necessity for task-specific training or fine-tuning, relies on providing the model with a language description or some task exemplars—referred to the *prompt*—that guide the desired autoregressive generation. Despite the remarkable success, the underlying mechanisms that facilitate such exceptional generalization abilities remain an open question. In this paper, we present a novel framework that formally conceptualizes answer generation for complex natural language tasks as a hierarchical "template-content" structure. According to our modeling, there exist pre-trained models that can automatically decompose tasks into constituent steps during autoregressive generation, through language modeling on a sufficiently large corpus, thereby solving them. Our framework offers an explanatory tool for the complex reasoning abilities of large language models from the perspective of modeling autoregressive generation tasks. Our experiments show that practical models exhibit different behaviors for "template" and "content" providing support for our modeling.

## 1 Introduction

The continuous evolution of pre-trained Large Language Models (LLMs) (Brown et al., 2020; Chowdhery et al., 2022; OpenAI, 2023) with ever-growing parameters and corpus sizes has notably augmented their capacity to solve complex tasks in natural language. These tasks range from arithmetic and symbolic logic to factual reasoning (Qin et al., 2023; Liu et al., 2023; Yang et al., 2022; Bang et al., 2023; Tan et al., 2023). With the prompts (Liu et al., 2021) (either exemplars (Brown et al., 2020) or task descriptions (Ouyang et al., 2022)), LLMs can solve the tasks through an autoregressive generation process, essentially following the prompts and creating the response token by token. The objective of transforming all processes into a *generative* format has elicited significant attention and curiosity regarding its mechanism. Within the reasoning community, a question of greatest concern is *"What mechanisms underlie the acquisition and application of LLMs' reasoning ability in autoregressive generation?"* Recently, several studies have been taken to explain how LLMs can accomplish reasoning on complex tasks from various perspectives. Akyurek et al. (2023) interpret the few-shot prompt as a learning algorithm that performs gradient descent on the linear model. Xie et al. (2022) provide an explanation from a Bayesian standpoint. However, their primary focus centers around in-context learning, and less on the autoregressive behavior.

In this paper, our primary objective is to draw a connection between autoregression and reasoning ability. To this end, we propose **template-content** structure, a modeling for natural language autoregressive generation, which divides the generation of sentences into two parts: the relatively task-specific **template** and the **content** that varies with specific questions. With this division, well-trained LLMs can learn the template as a flow indicator, which guides the LLMs to split a task into *sub-tasks*, a critical aspect of reasoning, during the autoregressive generation and then finish the given task by filling the template in accordance with the provided **content**. We provide an illustration in Figure 1. We further point out that the template-content structure could be **hierarchical** and **nested** in Section 5, which provides the models with the flexibility and capacity to combine various training

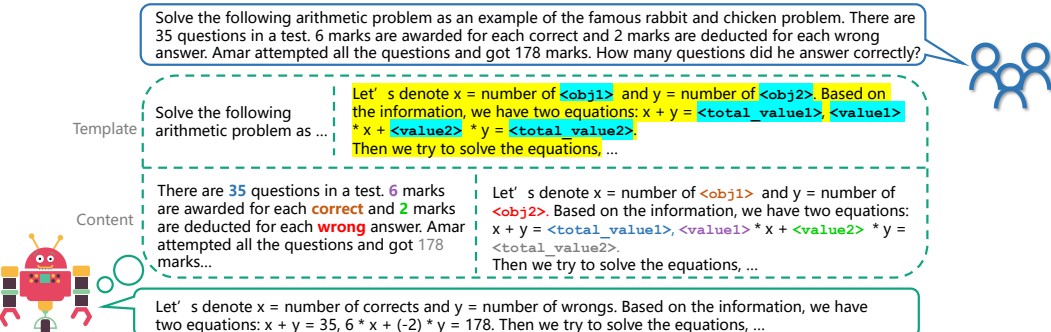

Figure 1: An illustration of the template-content structure. With the given prompt and question: 1. the model will generate the **template** tokens (highlighted as yellow) as a flow to solve the task according to the prompt, and some **content** placeholder (blue) in the template that needs to be filled in, which are displayed in the upper half of the dashed box. 2. The content generation with the guidance of the template such as <value> could be understood as **pointing**, shown in the bottom half. Here we use corresponding colors to show the pointing process. The combination of these two mechanisms ultimately leads to model output.

samples and, therefore, greatly enhances the generality of this structure. Our experiments provide evidence that current LLMs indeed demonstrate the template-content structure.

In addition, as a tool to support our claims, we recognize that prevailing research on model capacity has not been adequately extended to the currently most-used *causal* Transformer, i.e., the Transformer architecture with causal masked self-attention. Here and in the remaining part of the paper, the term "*causal*" means that the attention at the current token can only be allocated to the preceding tokens (Vaswani et al., 2017). As a side contribution to address the gap, we extend the well-known **universal approximation theorem** (UAT) for Transformers to models with causal masks in Section 3.

We hope our work can provide a new perspective and tool for understanding the reasoning ability of LLMs. Our contribution contains (1) We modify the UAT for Transformers to fit causal models, which is the most commonly used architecture in LLMs. (2) We propose a novel framework, i.e., the **template-content** structure to conceptualize the autoregressive generation for solving complex tasks. With our modification of the UAT, we show that the causal Transformer can implement the template-content structure. (3) We prove that if well-trained on a sufficiently large corpus, this Transformer can solve tasks by borrowing knowledge from its training data. By generalizing this structure to a hierarchical version, we can utilize its combinatorial capacity to explain the remarkable performance of large language models on complex tasks. (4) We conduct experiments to provide evidence that the current LLMs demonstrate the template-content structure.

## 2 BACKGROUND AND RELATED WORK

**Autoregressive language models** After the Transformer architecture (Vaswani et al., 2017) was proposed, there are two main research directions of the language models (Yang et al., 2023): one is the bi-direction encoder-only architecture, whose typical models are BERT (Kenton and Toutanova, 2019) and its following work RoBERTa (Liu et al., 2020), and another is the **causal** decoder-only architecture with the autoregressive generation, whose typical models are the series work of GPT (Radford et al., 2018; 2019; Brown et al., 2020). Specifically, these models employ *causal* attention layers. With the given beginning, the model generates the next token and repeats the process until a special stop token is generated, which is called *autoregressive* generation.

**Prompt engineering** The performance of autoregressive language models largely depends on the quality of prompts. The chain-of-thought (cot) prompt (Wei et al., 2022) shows LLMs have the step-by-step reasoning ability with exemplars in the same format and this task decomposition largely improve the performance. Even a simple "step-by-step" instruction can make a large progress (Kojima et al., 2022). The released chatGPT inherits the idea and forces the model to generate step-by-step answers during its RLHF phase (Ouyang et al., 2022). The basic idea of these prompts is to help or instruct models to split tasks into sub-tasks, which we believe is the key to reasoning ability.

**Capacity of LLMs** Two types of work have been undertaken on the capacity of LLMs. One of them is about the Turing completeness of the Transformers (Perez et al., 2021; Hahn, 2020) and another one focuses on the UAT, which is more related to our work. Yun et al. (2020) prove that the Transformer with positional embedding can approximate any continuous function with their construction. In our

paper, we will modify this theorem to show that the *causal* Transformer, can also approximate any *causal* continuous function (see Theorem 1). Following Yun et al. (2020), Luo et al. (2022) talk about the situation with relative position embedding and Haotian et al. (2023) give another proof of the UAT with more dimensions but fewer layers. Recently, Feng et al. (2023) follows the framework of Perez et al. (2021) and discusses the expressive capacity on some specific problems such as arithmetic tasks and dynamic programming in a more realistic situation, i.e., limited precision and context window.

**Explain the reasoning ability of LLMs**   There is some prior work on the topic: explaining the reasoning ability of LLMs. Most research focuses on the in-context learning ability, where models without finetuning learn from the given exemplars of the task. Akyurek et al. (2023) prove that transformers can implicitly implement gradient descent and closed-form algorithms for linear models using given exemplars. Xie et al. (2022) start from the long-range coherence during training and proposes that models learn potential document-level concepts and the in-context learning aims to learn the shared concept from the exemplars. Our framework, however, centers on task breakdown and template creation using autoregressive generation.

**Template and content**   Ford et al. (2018) classify words into "templates" and "contents" based on grammar or frequency and proposes a "two-pass" generation. They first generate "template" words and placeholders for "content" and then replace placeholders, which slightly enhances language modeling. Unlike them, we emphasize single-pass generation, highlighting the capacity of single-pass autoregressive models to distinct templates and contents.

## 3    UNIVERSAL APPROXIMATION THEOREM FOR CAUSAL MODEL

In this section, we will extend the UAT to the causal Transformer. The UAT proved in Yun et al. (2020) claims Transformers can approximate any continuous function (defined on a compact support set) with arbitrary precision. A formal formulation is in Appendix B.1 as Theorem 2. This theorem underscores the expressive capacity of Transformers and can serve as the foundation for explaining their reasoning mechanisms. However, to the best of our knowledge, there is currently no work that has extended this theorem to causal Transformers, which are the prevalent structures used in large-scale models today. Therefore, we first provide modifications to this theorem and then utilize it as a tool for studying the autoregressive model and template-content structure.

**Definition 1** (Causal sequence-to-sequence function).   The sequence-to-sequence function $f : \mathcal{D} \to \mathbb{R}^{n \times d}$ (where $\mathcal{D} \subseteq \mathbb{R}^{n \times d}$) is a causal function, if and only if for any two input $X$ and $X'$, and **any** $i \in [n]$, we have: $f(X)_{1:i} = f(X')_{1:i}$, if $X_{1:i} = X'_{1:i}$, where $X_{1:i}$ means the first $i$ rows of $X$. Because the first $i$ outputs only depend on the first $i$ inputs, we also denote the causal function as $f : \mathbb{R}^{i \times d} \to \mathbb{R}^{i \times d}$ for any $i \leq n$.

**Theorem 1** (Universal approximation theorem of causal Transformer).   *For any causal sequence-to-sequence function $f$ and the class of causal Transformer, which means all the attention is masked except for the preceding tokens, the convergence in Theorem 2 still holds.*

The proof of this theorem is roughly the same as the original theorem, except that there is a difference in a key bijective construction. We provide a highly simplified summary of the original proof as well as our construction in Appendix B.1.

## 4    TEMPLATE-CONTENT STRUCTURE

### 4.1   MOTIVATION

To understand why pre-trained autoregressive models handle various tasks, think of the two-step solving scheme: first sketching a basic outline (like a draft) and then filling in specifics. For example, when we solve math problems, we first decide the steps to take and then address each using given details, as depicted in Figure 1. This can be seen as having a general "template" and specific "contents". The template relates to the process of solving the overall task that is independent of the specific problem, while the content is specific to the particular problem.

By separating the template and content in an answer sequence, the process of generating an answer is divided into a relatively stable process (template) and a flexible process (content). This separation makes it possible to discuss how the models pre-trained on a large language corpus can **generalize** on reasoning task, where the word "generalize" means learning a question-independent task-solving

flow. If a model can separate the template and the content, we can expect it to learn the template from the huge corpus and then have the capacity to solve not only the same question that occurs in the training corpus but also different questions of the same task.

With a suitable template, the remaining work of the language model to fill in the details will be natural for LLMs. We describe the basic capacity of the content filling as *template-dependent pointing*. The generated template leaves blanks to be filled in with specific information, and also provides descriptions or "roles" for these blanks. For the example in Figure 1, with a template like "x = number of `<obj1>`", the left job for the content part is to find the corresponding object, property and value in the problem. So we describe it as *pointing*. It is not surprising that the language models have the ability, because it is also the basic ability to understand the semantics and finish many classic NLP tasks such as named entity recognition (Chiu and Nichols, 2016; Li et al., 2020) and translation. In a word, we believe learning to generate the template is key to solving complex tasks. Therefore, the focus of our theory will be on whether and how the model can generate such templates.

Above, we explain why the template-content structure can help us to understand the reasoning ability of LLMs. In the following, we will model the template-content structure in a formal way and use it to explain the reasoning ability of LLMs. We first model the autoregressive generation task with the template-content structure in Section 4.2 and then show (1) Transformer can model the template-content generation in Section 4.3 and (2) how a well-trained model can solve complex tasks with the template-content structure in Section 4.4.

## 4.2 TASK MODELING

Firstly, let us model the task we focus on. With a prompt and a question sequence as the start, a language model $\mathcal{M}$ generates an answer sequence autoregressively by continuing writing.

**Definition 2** (Prompt-leading autoregressive models for answer generation). For a task and the corresponding prompt sequence $\boldsymbol{p}$, a question $\boldsymbol{q}$ and a partial answer $\boldsymbol{a}_{1:t}$[1], $t \in \mathbb{N}$, all of which belong to $\mathcal{T}^*$ (the power set of the token space $\mathcal{T}$), we consider models $\mathcal{M}$ which generate the answer autoregressively (until generating the end of text token): $\mathcal{M}(\boldsymbol{p}, \boldsymbol{q}, \boldsymbol{a}_{1:t}) = a_{t+1} \in \mathcal{T}$.

As mentioned in Section 4.1, we believe the answer sequence includes the template part and content part. Here, we define the T/C classification function as follows:

**Definition 3** (Template and content, T/C). Function $\mathcal{F} : \mathbb{N} \times \mathcal{S} \to \{T, C\}$ is called a **T/C classification function**, where $\mathcal{S} \subseteq \mathcal{T}^*$.[2] The function takes a token sequence $\boldsymbol{a}$ and an index $i$ as input and gives the binary classification of the indexed token, denoted as $\mathcal{F}(i; \boldsymbol{a})$, abbreviated as $\mathcal{F}(a_i)$. We also use $\mathcal{F}(\boldsymbol{a})$ to denote the T/C sequence of the whole sequence.

To align with the autoregressive generation and causal model, in the following part, we always assume the T/C classification function (as a sequence-to-sequence function) should be a **causal** function. Figure 1 provides an intuition, where the yellow tokens are templates and blue are contents. We consider the distinct behavior between templates and content as an inherent characteristic of natural language, where the generation of templates is independent of specific contents, while contents relies on templates. We require this characteristic by introducing the definition of the **groundtruth** classification function and its corresponding **template-content** model (T-C model). For brevity, we place the formal definition in the Appendix B.2, and here we introduce its idea. The T-C model and the groundtruth classification defines such an **ideal** sequence generation schema: every token is classified into template or content, and for two sequences 1) when the preceding T/C sequences align and the template tokens at the corresponding positions are **the same**, then the classification of the next token to be generated is always **the same** (which is invariant to different content tokens in the preceding sequences), 2) furthermore, if the next token is a *template*, then the exact token is invariant to different content tokens in the preceding sequences, but only depend on the template tokens.

Additionally, we always consider prompts as templates and questions as contents. For instance, "the same template" requires the same prompt while the questions can be different. With some approximations, we believe that for natural language, such ideal modeling is appropriate, that is,

**Assumption 1** (Natural language has the template-content structure). There **exists** an autoregressive model $\mathcal{M}$ and a T/C classification function $\mathcal{F}$ that satisfy the requirements of the definition of the **T-C** model and the **groundtruth** classification.

---

[1] In the following, we use $\boldsymbol{a}_{1:t}$ to denote the first to $t$-th token sub-sequence or the empty sequence if $t = 0$.

[2] Here, we define this function only on a subset rather than the entire sequence set because we believe that this T/C classification is meaningful only in "normal" natural language rather than in random or garbled text.

The intuition behind this assumption is that we believe natural language inherently exhibits a hierarchical semantic phenomenon, allowing us to separate the more functional parts from the more specific parts, as mentioned in Section 4.1, demonstrated in Figure 1 and also talked in Ford et al. (2018). **Notice that** *here we use $\mathcal{M}$ to describe our ideal T-C-based autoregressive generation schema*, instead of *a specific parametric model or an algorithm*. We will prove that Transformer can implement the T-C model in Section 4.3 and give some empirical evidence that the natural language as well as real-world LLMs has this T-C structure in experiment (Section 6). We will also give a finer version with hierarchical templates in Section 5 and also some discussion such as generating a distribution instead of a token in Appendix C. Here, we focus on this binary setting for simplicity.

Until now, we have defined the natural language generation task as an autoregressive process in which template tokens only depend on preceding template tokens and content tokens can depend on both. This reflects our **core understanding** of natural language, where tokens can be classified into either "templates" or "contents". Template tokens define the flowing structure of the expression, and content tokens fill in the details. Template tokens are independent of content tokens, since changing the specific details should not alter the overall structure of the sentence. Below, we prove that the template-content structure (and the T-C model) can be achieved by a Transformer model.

## 4.3 THE EXISTENCE OF THE TEMPLATE-CONTENT TRANSFORMER

Given any preceding sequence $p, q, a_{1:t}$, a groundtruth classification function $\mathcal{F}$ and a T-C model $\mathcal{M}_I$, we want to prove there is a Transformer $\mathcal{M}$ that can generate the same tokens as $\mathcal{M}_I$.

The framework of our constructive proof is as follows. (1) We can reorganize the generation process of the T-C model $\mathcal{M}_I$: the input tokens are divided into two template tokens and content tokens according to the groundtruth T-C classification $\mathcal{F}$. Each group extracts its respective information $H_T$ and $H_C$ by a function $f_T$ and $f_C$, which is then combined by a function $g$ for the final output $a_{t+1}$. (2) Because of the UAT that we have proved in Section 3, the information extractor $f_T$, $f_C$ and the final output function $g$ can be all implemented by Transformers. (3) These three Transformers can be combined into a final T-C Transformer. Here we provide a diagram showing the architecture of our construction in Figure 2. The detailed constructive proof can be found in Appendix B.3.

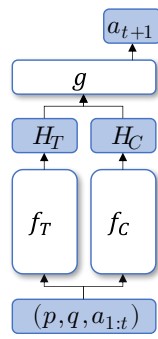

Figure 2: The architecture of the combined Transformer.

Until now, we have proven that there **exists** a Transformer that can generate following the template-content structure. In the rest, we always assume that the T/C classification function $\mathcal{F}$ is groundtruth and we use $(f_T, f_C, g)$ to represent a T-C Transformer constructed as above.

## 4.4 SAMPLE-BASED TEMPLATE GENERATION

In this section, we want to show that the template-content structure can explain how a pre-trained Transformer gains the ability to generate the template from a sample-based learning perspective, which leads to the task-solving capacity. Here we assume the pre-trained model perfectly fits the training samples.

**Definition 4** (Well-trained). A model $\mathcal{M}$ is *well-trained* on a training sample $(p, q, a)$, which means given the prompt $p$ and question $q$, the model can perfectly generate the answer $a$ autoregressively.

This definition requires a pre-trained model to "remember" and then reproduce a training example. We believe that such a requirement is not challenging for the prevailing LLMs with a huge amount of parameters, as it does not demand any form of generalization. Until now, we have demonstrated that (1) the template represents the general process of solving complex tasks in Section 4.1 and shown in Figure 1; (2) given our modeling of the autoregressive generation task using the T-C structure in Section 4.2, and our construction of the T-C Transformer in Section 4.3, the T-C Transformer exhibits the ideal behavior that enables template generation independent of specific content. Now, coupling the ideal behavior with the assumption that the Transformer is well-trained, it is a natural conclusion that this Transformer can generate the template according to training samples.

Here we describe the template generation as a partial sequence continuation problem and have the following proposition:

Table 1: A hierarchical template-content example, where the content is further decomposed into sub-template and sub-content. The different levels of the template are shown as underline ($T_1$), **bold** ($T_2$), and *italic* ($T_3$), as well as different indents in the answer.

| |
|---|
| [Prompt]: Solve the arithmetic problem step by step. *Melanie* **will be** *18* **years old in** *10* **years,** what is **the** *current age* **of** *Melanie*? |
| [Answer]: First, let's identify `<target value>`. According to the problem, `<information in question>`. This means that `<write in equation>`... |
|     `<target value>`: `<obj>`**'s** `<property>` 
         `<obj>`: *Melanie*, `<property>`: *current age* |
|     `<information in question>`: *Melanie* **will be** *18* **years old in** *10* **years** |
|     `<write in equation>`: `<variable>` **+** `<value1>` **=** `<value2>` 
         `<variable>`: *age*,  `<value1>`: *10*,  `<value2>`: *18* |
| [Final generation]: First, let's identify *Melanie***'s** *current age*. According to the problem, 
 *Melanie* **will be** *18* **years old in** *10* **years**. This means that *age* **+** *10* **=** *18*... |

**Proposition 1** (Answer generation). *With a pretrained T-C Transformer* $(f_T, f_C, g)$, *a prompt* $\boldsymbol{p}$, *question* $\boldsymbol{q}$ *and a prefix of a potential answer* $\boldsymbol{a}_{1:t}$ *(empty sequence if* $t = 0$*) as input, assuming that there exists a training sample* $(\boldsymbol{p}, \boldsymbol{q}', \boldsymbol{a}')$ *such that (1)* $\mathcal{F}(\boldsymbol{p}, \boldsymbol{q}, \boldsymbol{a}_{1:t}) = \mathcal{F}(\boldsymbol{p}, \boldsymbol{q}', \boldsymbol{a}'_{1:t})$, *(2)* $a_i = a'_i$ *for* $1 \leq i \leq t : \mathcal{F}(a_i) = T$ *and (3) the Transformer is well-trained on the sample, then the Transformer given the sequence* $(\boldsymbol{p}, \boldsymbol{q}, \boldsymbol{a}_{1:t})$ *can generate the answer* $\boldsymbol{a}$ *whose template tokens keep the* **same** *as* $\boldsymbol{a}'$, *i.e.,* $a_j = a'_j$ *for* $1 \leq j \leq |\boldsymbol{a}| : \mathcal{F}(a_j) = T$.

The proof is provided in Appendix B.4. This proposition conceptually demonstrates that, owing to our template-content modeling of natural language and the construction of the corresponding T-C Transformer, the Transformer can generate an answer sequence with the "correct" template as long as there is a training sample with the same template (but content may be different), thereby solving complex problems.

The only condition of the proposition is the existence of the training sample. The template is considered the **invariant part** for potentially infinite questions sharing the same task. The template space is much **smaller** than the total answer space, which is why we can assume the existence of a training sample with such a template that facilitates the LLM to invoke the same template when it sees a new question. This might explain the excellent **generalization** and **reasoning** abilities of modern LLMs. There are still some practical details such as our *token-wise alignment*, *prompt format*, *training process*, and *sampling strategy*, and we discuss them in Appendix C.

## 5 HIERARCHICAL TEMPLATE-CONTENT STRUCTURE

Above, we established the presence of the template-content Transformer and its capacity for template generation. However, we acknowledge that absolute binary T/C classification may be inadequate for more intricate real-world tasks. Tokens can vary in their level of specificity, aligning with diverse task segmentation granularity, ranging from the most general to the most specific. To describe the different specificity levels, we extend the template-content structure to the *hierarchical* and *nested* case. This extension entails content corresponding to a template at a given level being decomposable into **sub-template and sub-content** at the next level. Similarly, we no longer differentiate between prompt and question. In practical scenarios, a **holistic sequence** is often encountered, comprising **both** the prompt and the question intertwined, providing comprehensive information at various levels of detail. In the following, we uniformly call them "*prompt*", but assume tokens have hierarchical levels of meaning. An example is in Table 1, where we demonstrate the nesting relationship between different levels of content by progressively expanding the content into sub-template and sub-content. This presents a hierarchical structure from coarse- to fine-grained content generation.

The hierarchical structure can greatly generalize our framework by increasing its 1) **flexibility**: the template could range from general to specific, and 2) **completeness**: the structure allows the description of arbitrary complex (in)dependent logical relationships in sentences, given a sufficient nesting depth. Below, we formally describe the hierarchical T-C model.

### 5.1 HIERARCHICAL TASK MODELING

We define the hierarchical template-content structure as a multi-class classification: $\mathcal{F} : \mathbb{N} \times \mathcal{T}^* \rightarrow \{T_1, T_2, \ldots, T_n\}$. For simplicity, we denote the set $\{T_1, \ldots, T_k\}$ as $T_{\leq k}$, and $T_{\geq k}$ likewise. The $T_{k+1}$ can be seen as part of the content of $T_{\leq k}$ (lower-levels) as well as part of the template of $T_{\geq k+2}$ (higher levels). We also give tokens in the prompt the same hierarchical T/C classification, dividing the prompt into tokens of $\{T_1, T_2, \ldots, T_n\}$ and the different parts provide different levels

Figure 3: Sparsity of the dependency matrix. (a) The sequence is divided into 6 levels: $\boldsymbol{T_1}$, $T_2$, $T_3$, $\boldsymbol{T_4}$, $\boldsymbol{T_5}$, $\boldsymbol{T_6}$. (b) The dependency structure diagram for the full dependency of 6 levels templates. (c) The practical dependency structure for the sequence in (a), which is much sparser than (b).

of information to the corresponding levels of templates or content in the answer. For example, in Table 1, the prompt sequence "will be ... years old in ... years" ($T_2$) corresponds to the sub-template "... + ... = ..." ($T_2$) and sub-content "age", "10","18" ($T_3$) in the answer which together serve as a content `<write in equation>` of the parent template $T_1$. To enable a unified view and simpler notation, we merge the prompt into the beginning of the answer sequence $\boldsymbol{a}$.

The definition of groundtruth classification and T-C model parallels that in non-hierarchical setting. That is, identical T/C classification and $\leq k$-level template tokens yield identical T/C classification of the next position, including the token if it is a $T_{\leq k}$ token. The formal definition is presented in Appendix B.5. Similar to Section 4, subsequent mentions of the classification function $\mathcal{F}$ and model $\mathcal{M}$ are references to the groundtruth function and the T-C model respectively. The existence of the corresponding T-C Transformer $(f_{T_1}, \ldots, f_{T_n}, g)$ is the same as in the non-hierarchical situation and the proof is not reiterated here. The generating process of the T-C model (or the Transformer) can be represented as follows.

$$\boldsymbol{H}_{T_k,1:t} = f_{T_k}(\boldsymbol{a}_{1:t}), \quad k = 1, 2, \ldots, n, \qquad a_{t+1} = g\left(\boldsymbol{H}_{T_1,1:t}, \ldots, \boldsymbol{H}_{T_n,1:t}\right), \qquad (1)$$

where the function $f_{T_k}$ should only depend on the corresponding levels of tokens and $a_{t+1}$ should only depend on $\boldsymbol{H}_{T_1,1:t}, \ldots, \boldsymbol{H}_{T_k,1:t}$ if $\mathcal{F}(a_{t+1}) = T_k$.[3] In other words, a $T_k$ token $a_{t+1}$ only depends on preceding lower- or same-level tokens $T_{\leq k}$, but not on higher-level tokens $T_{\geq k+1}$.

**Sparse dependency in the hierarchical T-C model** In the hierarchical case, the dependence relationship could be refined. Notice that it is not necessary that the $k$-level tokens $T_k$ entirely depend on all the lower-level tokens $T_{\leq k-1}$. A small number of levels may be sufficient to provide enough information to determine the generation. An example is in Figure 3: solving an arithmetic problem in SingleEQ (Koncel-Kedziorski et al., 2015) dataset. In this example, the sequence (including the prompt and the generated answer) is segregated into six levels and its dependency is significantly sparser than the full dependency. For instance, the $T_5$ (brown) token "26" only depends on the $T_4$ (red) tokens "$54 - 28 =$", *regardless of* any information provided by other lower-level templates, such as where the equation appears ($T_1$) or the object bales ($T_3$) and so on. We call the phenomenon as **sparse dependency** and give it a formal definition in Appendix B.6. Dependency practically tends to be sparser than the full ones. This phenomenon prevents deeply nested T/C structures from imposing excessive constraints on higher levels (as in the case of full dependency, where the $T_n$ depends on all $n-1$ levels), enabling the model to learn deeper and more complex structures.

## 5.2 THE GENERALIZATION POWER OF THE HIERARCHICAL TEMPLATE-CONTENT MODELING

Generalizing the template-content structure to the hierarchical version can enable modeling arbitrarily complex tasks that may involve multiple levels of sub-tasks. We can get a similar but more general proposition with Proposition 1, which is the $n$-level hierarchical T-C models have the ability to generate answer from $n$ different samples, further explaining the generalization of the models. Considering space constraints, we defer the detailed discussion in the Appendix A. We define the *label* and *label consistency* as the conditions to combine different sentences in Appendix A.1 and give the formal discussion of the hierarchical answer generation ability in Appendix A.2. Here an informal description of the proposition is as follows:

**Proposition 2** (hierarchical answer generation, informal). *Given $n$ samples with label consistency on which a hierarchical T-C Transformer is well-trained, this model can generate the combined answer with the same $T_k$ tokens as the $k$-th sample.*

---

[3]The formal definition of "*depend on*" can be analogized similarly to what we did in Section 4.3, where the function always has the same output with identical T/C classification and $T_k$ tokens.

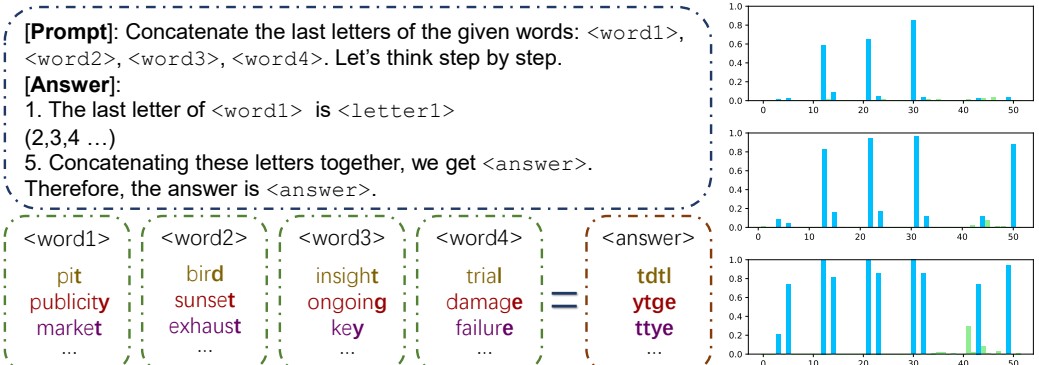

Figure 4: **Left**: The concatenate-last-letter dataset. The task is to concatenate the last letters of several words together. Here the template tokens are generated by GPT-4 and fixed, while all the content words (`<word>`, `<letter>`, `<answer>`) varies. **Right**: The variance of the output distributions at each position. X-axis: index of the tokens, y-axis: variance. **Ten** Blue bars: content tokens; Other green bars: template tokens. From top to bottom: GPT2-xl(1.5b), OPT-30b and Llama2-70b. Some green bars are too short to see.

# 6 EXPERIMENTS

Above, we propose the template-content structure framework and prove a T-C model (or Transformer) can achieve the generation with the T-C structure thereby solving complex tasks. In this section, we want to give some empirical evidence that the practical models exhibit similar behavior of our T-C models and therefore, our framework can be used to explain the reasoning ability of practical models.

## 6.1 VARIANCE OF THE OUTPUT

Reviewing the T-C model we defined in Section 4.2 and Appendix B.2, it requires an autoregressive model to identify the T/C classification of the next position given the preceding T-C classification and the template tokens. It should also generate results that are independent to content when the next position is template. To test if the real-word models align with it, we create a T-C dataset. Using GPT-4 (OpenAI, 2023) for the concatenate-last-letter task (Wei et al., 2022), we obtained an answer sequence. We labeled the *letters* and *words* as *content* and *the rest* as *template*. We use our manually-labeled T/C classification as the groundtruth. It is quite natural because of the relatively clear T/C classification in this task. After replacing content with other words and letters, we acquire sequences with aligned T/C classification and the same template tokens. In our experiments, we use more realistic *word*-level T/C alignment since different words can split into varying tokens by different models. We re-merging sub-word tokens and test on the first token for each word (details refers to Appendix D.3.1). This approach helps us compare sequences even when they have different token numbers and tokenizers. The dataset is illustrated in Figure 4 left with details are in Appendix D.1.

We input these sentences into various open-source models and measure the variance of the output distributions at each position. The output distribution is a vector with a dimension equal to the vocabulary size. We calculate the variance for each dimension and then average them. According to our definition, a **T-C** model should show **lower** variance for **template** tokens (given replacement only affects content, the template remains consistent for each input) and **higher** variance for **contents**.

The results of models GPT2-xl(1.5b: 1.5 billion parameters) (Radford et al., 2019), OPT-30b (Zhang et al., 2022) and Llama2-70b (Touvron et al., 2023) are displayed in Figure 4 right, where shorter template green bars compared to 10 content blue bars (4 words, 4 letters and 2 answers, see Figure 4 left) indicate significant less variance on the template positions. The results suggests that real-world models behave as the T-C model defines. In other word, our T-C structure can be applied to real-world models.

Interestingly, a model's ability to differentiate between T/C seems to correlate with its size and reasoning capabilities. The most powerful Llama2-70b with reasoning capabilities comparable to GPT-3.5, exhibits the clearest T/C distinction, while GPT2, struggling with this task with little CoT capacity (Kojima et al., 2022), also exhibits the least distinction. The results consistent with our theory: clear T/C distinction indicates better reasoning ability. And it also gives us confidence that (1) the ability to clearly differentiate T/C can serve as a criterion for judging a model's reasoning

Figure 5: The T/C classification generated by the autoregressive classifier based on a Llama-2-70b model. Template: yellow, content: blue. **Left**: concatenate-last-letter. **Right**: Arithmetic problem from SingleEQ. We mark the token whose classification conflicts with the human intuition as red.

capability and (2) our framework will likely fit future powerful LLMs. Further results with different templates and models are in Appendix D.2 as well as the AUC-ROC report.

## 6.2 Variance-based autoregressive T/C classifier

To better illustrate the differentiated behavior, we introduce an autoregressive T/C classifier based on the variance. The main method is similar to the previous section. However, unlike the experiments in Section 6.1 which requires manually labeling and write the content list for the whole sentences, we now only do so for the **prompt**. For any sentence, starting from the word right after the prompt, we iteratively predict the T/C classification word-by-word. For each position, by inputting the **preceding partial sentences** with several different content replacements, we measure the output variance at the current position. If the variance surpasses a predefined threshold, it is categorized as content, and we record the model's generation as the replacing tokens for it. Otherwise, we classify it as a template and directly add the original word to all perturbed sentences to ensure the same template. This process is repeated until the sentence is fully classified. A detailed pseudo-code is available in Appendix D.3.2. It is more practical as it eliminates the need to pre-label every sentence. But it is also challenging due to the autoregressive nature: any misclassification can negatively impact subsequent classifying. So we employ the powerful Llama-2-70b (Touvron et al., 2023).

We test a sentence derived from the aforementioned dataset (concatenate-last-letter task). The result is shown in Figure 5 left. For the concatenate-last-letter sentence where the structure of the sentence is clear, the result perfectly aligns with human intuition (which we believe is a very close approximation of the "groundtruth"), which means in this sentence, the Llama-2 model exhibits typical T-C behavior. To test on a more complex sentence, we choose a problem from an arithmetic dataset SingleEQ (Koncel-Kedziorski et al., 2015) and then generate an answer sequence by GPT-4, which leads to a longer sentence with a more complex structure than the above one from the concatenate-last-letter task. We choose the most typical content as $C$ including the names, objections, and Arabic numbers. The result is shown in Figure 5 right. The result is also consistent with our intuition at most positions even in this more complex setting, which supports our theoretical framework. Results on other sentences and the details of this classifier can be found in Appendix D.3.

## 7 Conclusions and Limitations

In this study, we propose a novel framework termed the template-content structure for modeling autoregressive generation in neural language. We demonstrated that with the template invariance, this framework can elucidate the ability of autoregressive models to tackle complex tasks. According to our construction, there exists a Transformer that can implement the generation with template-content structure and the natural corollary is its ability for complex reasoning. Furthermore, the hierarchical extension of our template-content framework explains the generalizing power of the models. We also provided some empirical evidence that practical models indeed exhibit distinct tendencies with regard to template and content separately.

There are still some limitations of this work. For our modeling, there is still some gap between our framework and the practical application, such as the alignment, sampling and so on. Despite some explanation and discussion in Appendix C, we acknowledge that further refinement of our framework is necessary, particularly in formalizing these discussions. This will be a pivotal focus of our future work. For our experiment, we acknowledge that our experiments rely on limited datasets and models. In addition, our experiments mostly focus on the output behavior, without more visualization of the inner behavior of these models. However, how to visualize the model behavior in a larger model is still an open question. This will also be our further work and we hope more visualization methods can help us to validate whether the models' behavior is consistent with our theoretical framework.

## REPRODUCIBILITY

We attach great importance to the reproducibility of our work. As a work that focuses on theoretical analysis, we give the assumptions and proofs of each theorem as formally and as detailedly as possible — the formal framework is the goal of our work in itself. Although limited by space and coherence, much of the more formal presentation has been added to the Appendix (see Appendix B). Regarding the algorithm used in the experiment, we describe in detail all the details including the generation of the data set and the implementation of the algorithm (see Appendix D). At the same time, we also provide codes in the supplementary materials, through which the experimental results in the article can be directly reproduced.

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

# A  HIERARCHICAL GENERATION

## A.1  HOW CAN WE COMBINE DIFFERENT SAMPLES INTO ONE SEQUENCE?

Generalizing the template-content structure to the hierarchical version can enable modeling arbitrarily complex tasks that may involve multiple levels of sub-tasks. However, the answer space also grows exponentially. Looking for a single training sample containing all levels of template guidance may not be realistic for the amount of data required, even for an Internet-scale training set. Fortunately, in this section, we show that it is feasible to combine different-level templates from different training samples. That is, we can learn a combinatorially complex hierarchical template from different training samples, each providing only a certain template, leading to also exponentially increased combinatorial power in the answer generation, which explains the generalization ability of our model.

To formally incorporate the "combination of different samples" into our T-C structure, we need to describe under what conditions these samples can be combined together to form a new sequence. We first define *label*. In Figure 1, we have already used some blue symbols such as `<obj1>`, `<obj2>`and `<value1>`to denote the **labels** for the content tokens, which then transforms into concrete tokens such as "`corrects`", "`wrongs`" and "`35`". We can think of a label as **sufficient** and **necessary** information from the template to generate the corresponding content, which means that any modification to the template without altering the label will not influence the generation of the content. For example, "`based on the formula <equ>`" and "`according to the equation <equ>`" are two template-content structures with different templates ("based on the formula" vs. "according to the equation") but the same label ("`<equ>`"). This label will generate exactly the same content (concrete equations) with the same preceding content information (the same arithmetic problem) for these two different templates. The formal definition is shown in Appendix B.7. If $a'$ is in the label set of $a$, we say the two sequences $a$ and $a'$ have *label consistency*. Intuitively, it means we can merge the template part of $a$ and the content part of $a'$ together to make a new sentence. And if these two sentences can be generated by an (ideal) T-C model separately, this combined sentence can be also generated by the model.

The concept of *label consistency* can be extended to $n$ samples and used to explain how the combination of $n$ samples can yield the hierarchical templates. Specifically, we have $n$ samples $a_1, \ldots, a_n$ with the aligned $n$-level T/C classification and want to merge them into one sequence by taking the $k$-th level tokens from the $k$-th sentences. When we have merged $T_{<k}$ levels from their respective samples, we can combine the $k$-th sample if the label of the $k$-th sample at the $k$-th level matches the combined sequence. We say these $n$ samples have *label consistency* if, for any $1 \leq k \leq n$, the label of $k$-th sample at the $k$-th level matches the combined sequence. In this situation, we denote the combined sequence as $\hat{a}$. Similar to the case of two sentences, this property ensures that as long as each sentence can be generated by a T-C model, the combined sentence can also be generated. The formal definition is shown in Appendix B.7.

## A.2  THE GENERALIZATION POWER OF THE HIERARCHICAL TEMPLATE-CONTENT MODELING

Above, the "*label*" and "*label consistency*" describe the conditions that several samples could be composed together into a new answer sequence. To achieve the combinatorial generation ability, another issue that needs to be addressed is **content generation**. Tokens can simultaneously serve as the content for lower-level tokens and the template for higher-level ones. The generation of tokens could either follow a content-like *pointing* approach or a template-like *continuing writing* strategy. For example in Figure 3, the token `54` and `28` should be produced by pointing whereas the resultant subsection `26` is likely learned from a training sample containing the same calculation. As we mentioned in Section 4.1, we focus on the template generating learned from training samples and assume the models have the content-generating ability. For the sake of brevity within our framework, we introduce the concept of *virtual training samples* to consolidate these two abilities. Provided a sequence can be autoregressively generated, meeting the criteria in Definition 4, we will treat it as a training sample. The sequence's actual presence in the training sample set, or its status as a virtual training sample capable of being generated because of assumed generalization ability (for example, generating content disparate from actual training samples), is inconsequential. This approach allows us to uniformly model answer generation within the sample-based continuous generation ability.

With our discussion of the condition that $n$ samples can be combined to provide corresponding-level templates, i.e., the label consistency, and the assumption about the content generation, we can now proceed to provide a proposition similar to Proposition 1 in the hierarchical structure.

**Proposition 3** (hierarchical answer generation, informal). *Given $k$ samples with label-consistency, which a T-C Transformer is well-trained on them, this model can generate the combined answer $\hat{a}$ with the same k-level tokens as the sample $a^{(k)}$.*

The formal description and the proof will be shown in Appendix B.8. The proposition demonstrates the model's capability to combine information from different training samples $a^{(i)}$. This capacity leads to exponentially increased combinatorial power in the answer generation as well as the generalization ability.

## B    FORMAL DEFINITION, PROPOSITION AND PROOF

### B.1    THE UNIVERSAL APPROXIMATION THEOREM

The formal description of the **original** UAT in Yun et al. (2020) is as follows:

**Theorem 2** (Universal approximation theorem, UAT). *For any continuous distribution $\mathcal{P}$ defined on a compact support $\mathcal{D} \subseteq \mathbb{R}^{n \times d}$, any $1 \leq p < +\infty$, $\varepsilon > 0$, the context window $n$ and the target continuous sequence-to-sequence function $f : \mathcal{D} \to \mathbb{R}^{n \times d}$, there exist a Transformer $g$ with $l$ layers, such that*

$$P_{\boldsymbol{X} \sim \mathcal{P}} \left[ \|f(\boldsymbol{X}) - g(\boldsymbol{X})\|_p < \varepsilon \right] \geq 1 - \varepsilon. \tag{2}$$

*where the norm is entry-wise $l_p$ norm.*[4]

The proof of our **modified** theorem is as follows.

*Proof.* Here, we follow the proof in Yun et al. (2020) (Theorem 3 in that paper). Notice that the only difference between the causal Transformer and the original Transformers is the mask in the self-attention. So we only need to modify the parts of the proof that related to self-attention.

The proof in Yun et al. (2020) can be divided into three steps:

1. Any continuous function defined on a compact support can be approximated by a piece-wise constant function on the $\delta$-grid $\mathbb{G}_\delta = \{0, \delta, \dots, 1 - \delta\}^{n \times d}$

2. Any piece-wise constant function can be approximated by a *modified* Transformers. Here, *modified* means that the softmax function in the self-attention is replaced by a hardmax function and the activation function can be any piece-wise linear function with at most three pieces.

3. The modified Transformers can be approximated by the original Transformers.

To modify the proof to the causal setting, the first step can be applied directly, as it is not contingent upon the specific structure of the Transformers. Similarly, the third step, which is proofed by approximating the hardmax through the softmax as the temperature approaches infinity, can be also applied directly. The only difference is the second step, the key part of the proof. The basic idea of the second step is several Transformer layers can be used to learn each input vector with its position and context to a unique representation like a *hash* function. Then, with the distinct representation and the universal approximating ability of feed-forward networks, expressive power can be achieved by the following feed-forward functions. Specifically, the part of the proof consists of three-step construction:

1. **Discretization**: A series of feed-forward layers in the modified Transformers network can quantize the continuous input $\boldsymbol{X} \in \mathbb{R}^{n \times d}$ into an element $\boldsymbol{L}$ on the extended grid $\mathbb{G}_\delta^+ = \{-\delta^{-nd}, 0, \delta, \dots, 1 - \delta\}^{n \times d}$. This step is to prepare the unique representation for the inputs.

---

[4]Here, we have slightly modified the form of the original theorem, such as the type of convergence.

2. **Unique Representation**: a series of self-attention layers can learn the *unique* representation $q(\boldsymbol{L}_i; \boldsymbol{L})$, where $\boldsymbol{L}_i$ is the input vector and $\boldsymbol{L}$ is the context. The representation is the same only if the input vector and the context are both the same.

3. **Value Mapping**: a series of feed-forward layers can map the unique representation to the desired output value.

The discretization and value mapping only involve the feed-forward layers and the proof can be applied directly. So we only need to modify the unique representation part. Specifically, modify the global context $\boldsymbol{L}$ to causal context $\boldsymbol{L}_{1:i}$ for the input $\boldsymbol{L}_i$.

As the claim in Appendix C in (Yun et al., 2020), the proof with position embedding only needs Category 1 in Appendix B.5.1. So we only need to modify this part to fit the causal setting. Here we claim:

1. With two additional dimensions in the hidden states to learn a position embedding, one self-attention layer with two (hardmax) heads can achieve the mapping from the *column id*[5] $l_k^{(t)}$ into the difference between the current position and the last position $\delta^{-2d}(l_k^{(t)} - l_{k-1}^{(t)})$

2. With $t$ stacks of such layers, the output at $t$-th position is the bijection mapping from the causal context $\boldsymbol{L}_{1:t}$.

The first claim: for the hardmax attention $\mathrm{Attn}_h(\boldsymbol{X}) = \sigma_H(\boldsymbol{X}\boldsymbol{W}_Q(\boldsymbol{X}\boldsymbol{W}_K)^T)(\boldsymbol{W}_V\boldsymbol{X})$.

First, we can use additional two dimensions in the hidden states to store the position embedding $(\cos(t\theta_n), \sin(t\theta_n))$ where $\theta_n = \frac{2\pi}{n}$. With the residual connection between each block, we just need to ensure the output of the attention blocks and feed-forward blocks in these dimensions are all zero so the position encoding will not change through different layers. We denote the extended input as $\boldsymbol{L}^+ \in \mathbb{G}_\delta^+ \times \mathbb{R}^{n \times 2}$.

Then let $\boldsymbol{W}_Q \in \mathbb{R}^{(d+2) \times 2} = (\boldsymbol{0}, \boldsymbol{R}(-\theta_n))$ where $\boldsymbol{R}$ is the rotation matrix in the 2-dimension plain and $\boldsymbol{W}_K \in \mathbb{R}^{(d+2) \times 2} = (\boldsymbol{0}, \boldsymbol{I}_2)$. So the $\boldsymbol{q}_t = (\cos((t-1)\theta_n), \sin((t-1)\theta_n))$ and $\boldsymbol{k}_t = (\cos(t\theta_n), \sin(t\theta_n))$ so that the hardmax at position $t$ always return the index $t-1$ (with additionally defining $\boldsymbol{v}_0 = \boldsymbol{v}_1$). As for $\boldsymbol{W}_v$, we just use the construction in the original proof, which means $\boldsymbol{W}_v \in \mathbb{R}^{(d+2) \times 1} = (1, \delta^{-1}, \ldots, \delta^{-d+1}, 0, 0)$. This head returns the *column index* $l_{t-1}$ at the position $t-1$. Let another head returns $l_t$ and the $W_h = \delta^{-2d}(1, -1)$, so that after one self-attention layer, the value $\boldsymbol{v}_t$ at position $t$ is $l_t + \delta^{-2d}(l_t - l_{t-1})$. Then we repeat the layer $n$ times, we can easily prove that the value $\boldsymbol{v}_t$ at position $t$ is

$$l_t^{(n)} = \sum_{i=0}^{n} \delta^{-2id} \sum_{k=0}^{i} \left( \binom{i}{k}(-1)^k l_{t-k} \right), \tag{3}$$

where we use the convention that $l_t = l_1$ if $t \leq 0$. Now let us show why the value $l_t^{(n)}$ is the bijection mapping from the causal context $\boldsymbol{L}_{1:t}$. Note that $|\sum_{k=0}^{i}(\binom{i}{k}(-1)^k l_{t-k})| \leq 2^i(\delta^{-d+1} - \delta) \leq (\delta^{-d} - 1)$ if we set $\delta \leq 1/2^n$. So if we have $l_t^{(n)} = l_t^{(n)'}$, denote $c_i = \sum_{k=0}^{i} (\binom{i}{k}(-1)^k l_{t-k})$, then we must have

$$\sum_{i=0}^{n} \delta^{-2id}(c_i - c_i') = 0, \quad \text{where } -(\delta^{-d} - 1) \leq c_i \leq (\delta^{-d} - 1), \tag{4}$$

and

$$|c_i - c_i'| \geq \delta \text{ or } c_i = c_i'. \tag{5}$$

because the each $l_i$ differ at least $\delta$ after the discretization and with the special construction of $\boldsymbol{W}_v$.

If $c_n \neq c_n'$, we have

$$|\sum_{i=0}^{n-1} \delta^{-2id}(c_i - c_i')| \leq \sum_{i=0}^{n-1} 2\delta^{-2id}(\delta^{-d} - 1) \leq 2(\delta^{-2nd} - 1)/(\delta^{-d} + 1) < \delta^{-2nd+1} \leq \delta^{-2nd}|c_i - c_i'|, \tag{6}$$

---

[5]The column id is a unique representation of the input vector. See Appendix B.5 of the original paper.

but at the same time,

$$|\sum_{i=0}^{n-1} \delta^{-2id}(c_i - c_i')| = |\delta^{-2nd}(c_i - c_i')| = \delta^{-2nd}|c_i - c_i'|, \tag{7}$$

because of the sum in Equation (4) is zero, there is a conflict. So we must have $c_n = c_n'$. And similarly, we can recursively prove each $c_i$ and $c_i'$ are equal. Note that $c_0 = c_0'$ implies $l_t = l_t'$, and $c_1 = c_1'$ implies $(l_t - l_{t-1}) = (l_t' - l_{t-1}')$ so that $l_{t-1} = l_{t-1}'$. We can recursively prove that $l_i = l_i'$ for any $i = 1, \dots, t$. Combining with the bijection of $\boldsymbol{L}_i$ to $l_t$ (proofed in the original paper), it means the mapping from $\boldsymbol{L}_{1:t}$ to $l_t^{(n)}$ is a bijection. Because our results are still bounded, the properties 6.3 and 6.4 can be satisfied with slight modification. This finishes the proof. □

### B.2 THE GROUNDTRUTH CLASSIFICATION AND THE TEMPLATE-CONTENT GENERATION MODEL

**Definition 5** (The groundtruth classification and the template-content generation model, formal)**.** If a T/C classification function $\mathcal{F}$ and an autoregressive generation model $\mathcal{M}$ satisfy the following requirements, we call the function $\mathcal{F}$ as a **groundtruth** T/C classification and the model $\mathcal{M}$ as a **template-content (T-C)** model. The requirements are: for any prompt $\boldsymbol{p}$, question $\boldsymbol{q}$ and $\boldsymbol{q}'$, partial answer $\boldsymbol{a}_{1:t}$ and $\boldsymbol{a}_{1:t}'$, (1) if $\mathcal{F}(\boldsymbol{p}, \boldsymbol{q}, \boldsymbol{a}_{1:t}) = \mathcal{F}(\boldsymbol{p}, \boldsymbol{q}', \boldsymbol{a}_{1:t}')$ (T/C alignment) and $a_s = a_s'$ for all $1 \le s \le t$ such that $\mathcal{F}(a_s) = T$ (the same template), then the T/C classification of the next token is the same:

$$\mathcal{F}\left(\mathcal{M}(\boldsymbol{p}, \boldsymbol{q}, \boldsymbol{a}_{1:t})\right) = \mathcal{F}\left(\mathcal{M}(\boldsymbol{p}, \boldsymbol{q}', \boldsymbol{a}_{1:t}')\right), \tag{8a}$$

and (2) if $\mathcal{F}(\boldsymbol{p}, \boldsymbol{q}, \boldsymbol{a}_{1:t}) = \mathcal{F}(\boldsymbol{p}, \boldsymbol{q}', \boldsymbol{a}_{1:t}')$ (T/C alignment), $a_s = a_s'$ for all $1 \le s \le t$ such that $\mathcal{F}(a_s) = T$ (the same template), and **further** $\mathcal{F}\left(\mathcal{M}(\boldsymbol{p}, \boldsymbol{q}, \boldsymbol{a}_{1:t})\right) = \mathcal{F}\left(\mathcal{M}(\boldsymbol{p}, \boldsymbol{q}', \boldsymbol{a}_{1:t}')\right) = T$, then

$$\mathcal{M}(\boldsymbol{p}, \boldsymbol{q}, \boldsymbol{a}_{1:t}) = \mathcal{M}(\boldsymbol{p}, \boldsymbol{q}', \boldsymbol{a}_{1:t}'). \tag{8b}$$

### B.3 THE EXISTENCE OF THE T-C TRANSFORMERS

Here, we consider an ideal T-C model $\mathcal{M}_I$ and the corresponding groundtruth T/C classification $\mathcal{F}$. We need to construct a Transformer to fit the model $\mathcal{M}_I$.

We first rewrite the generation process of the T-C model $\mathcal{M}_I$ in Equation (**??**) as follows:

$$a_{t+1} = \mathcal{M}_I(\boldsymbol{p}, \boldsymbol{q}, \boldsymbol{a}_{1:t}) = g_I(f_{T,I}(\boldsymbol{p}, \boldsymbol{q}, \boldsymbol{a}_{1:t}), f_{C,I}(\boldsymbol{p}, \boldsymbol{q}, \boldsymbol{a}_{1:t})) \tag{9}$$

where $f_{T,I}$ and $f_{C,I}$ are causal functions that map an input sequence to a sequence with the same length in a certain representation space, and $g_I$ is a causal function to output the next token prediction. The $f_{T,I}$ is the function to extract the template information independent of any content token. Specifically, it means $f_{T,I}(\boldsymbol{p}, \boldsymbol{q}, \boldsymbol{a}_{1:t}) = f_{T,I}(\boldsymbol{p}, \boldsymbol{q}', \boldsymbol{a}_{1:t}')$ if $\mathcal{F}(\boldsymbol{p}, \boldsymbol{q}, \boldsymbol{a}_{1:t}) = \mathcal{F}(\boldsymbol{p}, \boldsymbol{q}', \boldsymbol{a}_{1:t}')$ (T/C alignment), and $a_s = a_s'$, $\forall 1 \le s \le t : \mathcal{F}(a_s) = T$ (same template). The $f_{C,I}$ is similarly defined as the content representation only depending on the preceding content tokens and being independent of any template token. That is, $f_{C,I}(\boldsymbol{p}, \boldsymbol{q}, \boldsymbol{a}_{1:t}) = f_{C,I}(\boldsymbol{p}', \boldsymbol{q}, \boldsymbol{a}_{1:t}')$ if $\mathcal{F}(\boldsymbol{p}, \boldsymbol{q}, \boldsymbol{a}_{1:t}) = \mathcal{F}(\boldsymbol{p}', \boldsymbol{q}, \boldsymbol{a}_{1:t}')$ (T/C alignment), and $a_s = a_s'$, $\forall 1 \le s \le t : \mathcal{F}(a_s) = C$ (same content). To ensure that template token only depends on preceding template tokens, we also have when $\mathcal{F}(a_{t+1}) = T$, the value of function $g_I$ should only depend on the output of $f_{T,I}$ while being invariant to changes of $f_{C,I}$.

Considering that the vocabulary is finite, we can set the representation space of $f_{T,I}$ and $f_{C,I}$ as $\mathbb{R}^d$ where for a sufficiently large $d$, the capacity is guaranteed to be adequate. In this situation, the value of $f_{T,I}$ and $f_{C,I}$ are two tensors shaped as $(|\boldsymbol{p}| + |\boldsymbol{q}| + t) \times d$ (denoted as $\boldsymbol{H}_{T,I}$ and $\boldsymbol{H}_{C,I}$), and we can concatenate these two tensors in the second dimension as the input of $g_I$. Notice that we can require $f_{T,I}$, $f_{C,I}$ and $g_I$ to be causal functions[6] because of the autoregressive generation. According to Theorem 1, there **exist** three causal Transformers that can approximate these function $f_{T,I}$, $f_{C,I}$ and $g_I$ arbitrarily well and we denote them as $f_T$, $f_C$ and $g$.

---

[6]To be more precise, we can construct a function $\hat{g}_I$ that generates a sequence $(\boldsymbol{p}_{2:|\boldsymbol{p}|}, \boldsymbol{q}, \boldsymbol{a}_{1:t+1})$, where this function is causal. Then, the $g_I$ takes the last element of the output of $\hat{g}$.

Finally, we need to show that these three Transformers can be combined into one (through concatenation and stacking). By combining the heads in each layer of $f_T$ and $f_C$ together, the output of the combined layer is the concatenation of the output of $f_T$ and $f_C$. We can keep their independence property by ensuring that the feed-forward layer is block diagonal. As in other theoretical analyses, we ignore the layer norm. After $l = \max(\text{num\_layer}(f_T), \text{num\_layer}(f_C))$ layers, the output of the combined Transformer is the concatenation of $\boldsymbol{H}_T$ and $\boldsymbol{H}_C$. Then it just needs to add a Transformer $g$ after the combined $f_T$ and $f_C$. As a summary, the constructed Transformers is as shown in Figure 2. Namely, $f_T, f_C$ compose the first few layers of the combined Transformer, their output tensors $\boldsymbol{H}_T, \boldsymbol{H}_C$ are the hidden representations after these layers, and finally $g$ takes these hidden representations to output the next token.

### B.4 THE PROOF OF PROPOSITION 1

The proof of Proposition 1 is shown as follows:

*Proof.* Given the prefix $\boldsymbol{a}_{1:t}$ and $\boldsymbol{a}'_{1:t}$, we considering the generation of the next token $a_{t+1}$. Because the Transformer is well-trained on the training sample $\boldsymbol{p}, \boldsymbol{q}', \boldsymbol{a}'$, according to the Definition 4 (well-trained model), we have

$$a'_{t+1} = \mathcal{M}(\boldsymbol{p}, \boldsymbol{q}', \boldsymbol{a}'_{1:t}), \tag{10}$$

We denote $\mathcal{M}(\boldsymbol{p}, \boldsymbol{q}, \boldsymbol{a}_{1:t})$ as $a_{t+1}$. According to the Equation (8a), we have $\mathcal{F}(a_{t+1}) = \mathcal{F}(a'_{t+1})$. If the generated token is a content token, it has satisfied the requirement of the proposition, because the proposition only claims the same template tokens. If the generated token is a template token, according to the Equation (8b), we have

$$a_{t+1} = \mathcal{M}(\boldsymbol{p}, \boldsymbol{q}, \boldsymbol{a}_{1:t}) = \mathcal{M}(\boldsymbol{p}, \boldsymbol{q}', \boldsymbol{a}'_{1:t}) = a'_{t+1}, \tag{11}$$

which satisfies the claim. Finally, we need to check whether the conditions of the proposition have been satisfied for the new sequence concatenated with the generated token $a_{t+1}$. If the conditions are still satisfied, we can finish the proof recursively. For the first condition (T/C alignment), we have $\mathcal{F}(a_{t+1}) = \mathcal{F}(a'_{t+1})$ and $\mathcal{F}(\boldsymbol{p}, \boldsymbol{q}, \boldsymbol{a}_{1:t}) = \mathcal{F}(\boldsymbol{p}, \boldsymbol{q}', \boldsymbol{a}'_{1:t})$. Because the function $\mathcal{F}$ is a causal function and the newly appended token does not affect the preceding value, we have $\mathcal{F}(\boldsymbol{p}, \boldsymbol{q}, \boldsymbol{a}_{1:t+1}) = \mathcal{F}(\boldsymbol{p}, \boldsymbol{q}', \boldsymbol{a}'_{1:t+1})$. The second condition (the same template tokens) is obviously satisfied. □

### B.5 THE FORMAL DEFINITION OF HIERARCHICAL TEMPLATE INDEPENDENCE

Here, we give the generalization of the Definition 5 in the hierarchical situation. First, we need to generalize the conditions as follows:

**Definition 6** ($k$-th level alignment). With a $n$ level hierarchical classification function $\mathcal{F}$, two sequences $\boldsymbol{a}$ and $\boldsymbol{a}'$ with same length are called *$k$-th level aligned*, if and only if for any $1 \leq t \leq |\boldsymbol{a}|$, at least one of the following conditions is satisfied:

$$\mathcal{F}(a_t) = \mathcal{F}(a'_t) \text{ and } a_t = a'_t, \tag{12}$$

or

$$\mathcal{F}(a_t) \in T_{\geq k+1} \text{ and } \mathcal{F}(a'_t) \in T_{\geq k+1}. \tag{13}$$

To show that the definition is the generalization about the T/C alignment and fixed template tokens, notice that if we set $n = 2$ and $k = 1$, the second condition says that both the tokens are content tokens while the first condition says that both are template tokens and the same. In this situation, the condition is the same as the ones that we use in Definition 5 and Proposition 1, i.e., the T/C classification is aligned and the template tokens are the same. Then, the generalization of the Definition 5 is as follows:

**Definition 7** (The groundtruth classification and the template-content generation model, hierarchical). For a hierarchical T/C classification function $\mathcal{F}$ with $n$ level and an autoregressive generation model $\mathcal{M}$, for any $1 \leq k \leq n$, given the *$k$-th level aligned* prefix sequences $\boldsymbol{a}_{1:t-1}$ and $\boldsymbol{a}'_{1:t-1}$, if the generated (next) token is also $k$-th level aligned during the autoregressive generation, we call the function $\mathcal{F}$ as a **groundtruth** T/C classification and the model $\mathcal{M}$ is a T-C model. It means for all sequences $\boldsymbol{a}$ and $\boldsymbol{a}'$ with the same length, and for all $1 \leq t \leq |\boldsymbol{a}|$, **at least one** of the following equations should be satisfied:

$$\mathcal{F}\left(\mathcal{M}(\boldsymbol{a}_{1:t-1})\right) = \mathcal{F}\left(\mathcal{M}(\boldsymbol{a}'_{1:t-1})\right) \text{ and } \mathcal{M}(\boldsymbol{a}_{1:t-1}) = \mathcal{M}(\boldsymbol{a}'_{1:t-1}), \tag{14}$$

**or**

$$\mathcal{F}\left(\mathcal{M}(\boldsymbol{a}_{1:t-1})\right) \in T_{\geq k+1}, \text{ and } \mathcal{F}\left(\mathcal{M}(\boldsymbol{a}'_{1:t-1})\right) \in T_{\geq k+1}, \qquad (15)$$

if the prefix sequences $\boldsymbol{a}_{1:t-1}$ and $\boldsymbol{a}'_{1:t-1}$ are $k$-th level aligned.

## B.6 THE FORMAL DEFINITION OF DEPENDENCY MATRIX

Formally, for a given sentence $\boldsymbol{a}$, when the next token is $T_k$, there is a **support set** $\boldsymbol{D}_k \in \{0,1\}^k$ such that

$$g(f_{T_1}(\boldsymbol{a}), \dots, f_{T_k}(\boldsymbol{a})) = g(f_{T_1}(\boldsymbol{a}^{(1)}), \dots, f_{T_k}(\boldsymbol{a}^{(k)})), \forall \boldsymbol{a}^{(s)} \in \mathcal{S}^{(s)}, s = 1, 2, \dots, k,$$

where $\mathcal{S}^{(s)}$ is a one-point set $\{\boldsymbol{a}\}$ if the $s$-th element of $\boldsymbol{D}_k$ (denoted as $d_{ks}$) is 1, otherwise $\mathcal{S}^{(s)}$ is the full space of the sequence with the same length. In this situation, we refer to the $k$-th level as (conditional) **independent** of the $s$-th levels if $d_{ks} = 0$. The support set must exist because if we set it as $(1, 1, \dots, 1)$, it says nothing about the function $g$. Then We define the **dependency matrix** as a lower-triangle matrix where $\boldsymbol{D}_{k,1:k} = \boldsymbol{D}_k$.

## B.7 THE FORMAL DEFINITION OF LABEL

Formally, we define

**Definition 8** (Label). For the given model $\mathcal{M} = (f_{T_1}, \dots, f_{T_n}, g)$, a sequence $\boldsymbol{a}_{1:t}$ and its dependency matrix $\boldsymbol{D}$, when the next token to be generated $a_{t+1}$ satisfies $\mathcal{F}(a_{t+1}) = T_k$, the $k$-th level label of $\boldsymbol{a}_{1:t}$ (denoted as $\mathcal{L}_k(\boldsymbol{a}_{1:t})$) is defined as the set of sequences $\{\boldsymbol{a}'_{1:t}\}$:

1. The T/C classification is aligned, i.e., $\mathcal{F}(\boldsymbol{a}'_{1:t}) = \mathcal{F}(\boldsymbol{a}_{1:t})$.

2. The combined sequence $\hat{\boldsymbol{a}}_{1:t}$ is constructed by replacing all the $\leq (k-1)$-level tokens in $\boldsymbol{a}_{1:t}$ with those in $\boldsymbol{a}'_{1:t}$. The replacement does not affect

   (a) the T/C classification of the next token, which means

   $$\mathcal{F}(\mathcal{M}(\hat{\boldsymbol{a}}_{1:t})) = \mathcal{F}(\mathcal{M}(\boldsymbol{a}_{1:t}))$$

   (b) the generation of the next token, which means

   $$g(f_{T_1}(\boldsymbol{a}'_{1:t}), \dots, f_{T_{k-1}}(\boldsymbol{a}'_{1:t}), f_{T_k}(\boldsymbol{a}_{1:t})) = g(f_{T_1}(\boldsymbol{a}_{1:t}), \dots, f_{T_k}(\boldsymbol{a}_{1:t}));$$

   (c) the dependency matrix, which means the dependency matrix $\boldsymbol{D}$ of the sequence $\boldsymbol{a}_{1:t}$ is also the dependency matrix of sequence $\hat{\boldsymbol{a}}$.

With the statement 2a, 2b in the definition, we can rewrite the generation as $a_{t+1} = g(f_{T_1}(\mathcal{L}_k(\boldsymbol{a}_{1:t})), \dots, f_{T_{k-1}}(\mathcal{L}_k(\boldsymbol{a}_{1:t})), f_{T_k}(\boldsymbol{a}_{1:t}))$, if $\mathcal{F}(a_{t+1}) = T_k$. It is worth noting the similarity between this definition and the definition of T-C model (Definition 5 and 7 in Appendix). To ensure the same next token generation, the "having the same label" condition is a relaxation of the "having the same $\leq T_{k-1}$ token" condition, whereby the exact template can be substituted with equivalent sequences. For the sequence $\boldsymbol{a}'$ in the label set $\mathcal{L}_k(\boldsymbol{a})$, the information from the dependent-level template tokens of $\boldsymbol{a}'$ is equivalent to those of $\boldsymbol{a}$. It enables the possibility of combining two sequences from different sources during the "continuation" generation of the T-C model. That is, the $k$-level template part from the original sequence $\boldsymbol{a}$ can be combined with the lower-level template part from another sequence $\boldsymbol{a}'$, which does not impact the generation of $T_k$ tokens, just like we can combine the template "`according to the equation`" with the content in "`based on the formula <equ>`" to generate an appropriate sentence "according to the equation <equ>".

With the statement 2c, it further claims the replacement does not introduce additional inter-level dependency. When the support set describes the dependency of the original sequence $\boldsymbol{a}$, the sequence can be replaced solely on these dependent levels, while the arbitrariness of other independent levels can still be maintained. For example, when we replace "`based on the formula <equ>`" with "`according to the equation`", the replacing sequence keeps the independence of further lower-level templates and therefore we can combine them.

We formally define *label consistency* as follows:

**Definition 9** (Label consistency). For a set of sequences $\{\boldsymbol{a}^{(k)} \mid k = 1, \ldots, n\}$ with $n$ levels of template, they have *label consistency* if and only if the following requirements are satisfied.

1. The T/C classification of these sequences is aligned.

$$\mathcal{F}(\boldsymbol{a}^{(i)}) = \mathcal{F}(\boldsymbol{a}^{(j)}), \quad \forall i, j \in \{1, \ldots, n\}. \tag{16a}$$

2. We construct the **combined sequence** $\hat{a}$ which takes the $T_k$ tokens from the corresponding sequence $\boldsymbol{a}^{(k)}$, i.e., $\hat{a}_t = a_t^{(k)}$ if $\mathcal{F}(a_t^{(i)}) = T_k, \forall i$. For any $0 \leq t \leq |\hat{a}| - 1$, and $\mathcal{F}(\hat{a}_{t+1}) = T_k$,

$$\hat{\boldsymbol{a}}_{1:t} \in \mathcal{L}_k(\boldsymbol{a}_{1:t}^{(k)}). \tag{16b}$$

The first requirement (Equation (16a)) serves the purpose of constructing the combined sequence $\hat{a}$. As the key requirement, the second one (Equation (16b)) ensures that the $k$-th sample can be merged with the 1st to $(k-1)$-th samples "*appropriately*", collectively forming the $k$-th template, by requiring its $k$-th level label should match the combined sequence $\hat{a}$. Here, when we say "appropriately", intuitively, it means that the combined sentence remains coherent and reasonable and, thus, for an ideal T-C autoregressive model, it can still generate the same $k$-level tokens. At the same time, this requirement also determines the dependency matrix of the sequence $\hat{a}$. Specifically, the $k$-th row of the dependency matrix is the same as the $k$-th row of the dependency matrix for the $k$-th sample.

### B.8 THE PROOF OF PROPOSITION 3

The formal description of the Proposition 3.

**Proposition 4** (Hierarchical answer generation, formal). *Given a partial answer $\boldsymbol{a}_{1:t}$ as the input and an $n$-level hierarchical template-content classification function $\mathcal{F}$, and a T-C Transformer model $(f_{T_1}, \ldots, f_{T_n}, g)$, we assume that there exist a set of training samples $\{\boldsymbol{a}^{(k)} | k = 1, \ldots, n\}$ that has label consistency (with denoting the combined sequence as $\hat{\boldsymbol{a}}$), and the partial sequence $\hat{\boldsymbol{a}}_{1:t} = \boldsymbol{a}_{1:t}$ and the model $\mathcal{M}$ is well-trained on them. We have the model can generate the answer $\boldsymbol{a}$ from $\boldsymbol{a}_{1:t}$ autoregressively as the same tokens as the combined sequence $\hat{\boldsymbol{a}}$, i.e., with the same $k$-level tokens as the training sample $\boldsymbol{a}^{(k)}$ for any $k$ from 1 to $n$. Here, we require that each prompt of each sample $\boldsymbol{p}^{(k)}$ should be contained in the partial sequence $\boldsymbol{a}_{1:t}$ or be generated as a part of $T_{\leq k-1}$.*

*Proof.* Here, we prove that if the assumption $\hat{\boldsymbol{a}}_{1:t+s} = \boldsymbol{a}_{t+s}$ holds for any the partial answer $a_{t+s}$ where $s$ takes value from 0 to $l - t - 1$, then $\hat{\boldsymbol{a}}_{1:t+s+1} = \boldsymbol{a}_{t+s+1}$ also holds, where we use $a_{t+s}$ to denote the generated partial sequence with length $t + s$.

Without loss of generality, we assume the next token $\hat{a}_{1:t+s+1}$ is a $T_k$ token. Because the training samples have label consistency, we have the T/C classification of the generating token $a_{1:t+s+1}$ is also a $T_k$ token, i.e.,

$$\mathcal{F}(a_{t+s+1}) = \mathcal{F}(\hat{a}_{t+s+1}) = T_k \tag{17}$$

Therefore, the token can be generated by

$$a_{t+s+1} = g(f_{T_1}(\boldsymbol{a}_{1:t+s}), \ldots, f_{T_k}(\boldsymbol{a}_{1:t+s})). \tag{18}$$

And we have the assumption $\hat{\boldsymbol{a}}_{1:t+s} = \boldsymbol{a}_{t+s}$, so we have

$$a_{t+s+1} = g(f_{T_1}(\hat{\boldsymbol{a}}_{1:t+s}), \ldots, f_{T_k}(\hat{\boldsymbol{a}}_{1:t+s})). \tag{19}$$

Because the label consistency, we have $\hat{\boldsymbol{a}}_{1:t+s} \in \mathcal{L}_k(\boldsymbol{a}_{1:t+s}^{(k)})$ and notice that $f_{T_k}(\boldsymbol{a}_{1:t+s}^{(k)}) = f_{T_k}(\hat{\boldsymbol{a}}_{1:t+s})$ because the combined sequence has the same $T_k$ tokens as the $k$-th sample $\boldsymbol{a}^{(k)}$, so we have

$$g(f_{T_1}(\hat{\boldsymbol{a}}_{1:t+s}), \ldots, f_{T_k}(\hat{\boldsymbol{a}}_{1:t+s})) = g(f_{T_1}(\boldsymbol{a}_{1:t+s}^{(k)}), \ldots, f_{T_k}(\boldsymbol{a}_{1:t+s}^{(k)})). \tag{20}$$

At the same time, according to the assumption of well-training and the prompt, we have that the model can generate the token $a_{t+s+1}^{(k)}$ given the sequence $\boldsymbol{a}_{1:t+s}^{(k)}$, so we have

$$a_{t+s+1} = a_{t+s+1}^{(k)}. \tag{21}$$

According to the definition of the combined sequence $\hat{a}$, $\hat{a}_{t+s+1}$ takes value from $a_{t+s+1}^{(k)}$. So we finally prove that $a_{t+s+1} = \hat{a}_{t+s+1}$. $\square$

## C    DISCUSSION OF THE TEMPLATE-CONTENT FRAMEWORK

*Remark* 1 (Practical training process). In our assumption, training samples should be in the format as the (prompt, question, answer) triplets while the Internet corpus may not be primarily presented in such a format. We point out that this alignment could be achieved in the crucial finetuning process i.e. RLHF or similar finetuning phase. In this phase, the model can realign previously learned corpus into the triplet format.

*Remark* 2 (format of the prompt). There are two primary prompt types: zero-shot instruction and few-shot exemplars and our template-content structure is applicable to both. Zero-shot prompts aid the model in recalling the templates learned during training, while few-shot prompts can also provide explicit templates. By utilizing these few-shot exemplars, models can generate templates based on the exemplars and also leverage the knowledge from similar templates, while the content information should not be directly used in answer generation. This observation also explains why zero-shot learning is more challenging than few-shot learning, as zero-shot learning necessitates the model to independently generate templates. This concept aligns with the findings in Min et al. (2022), demonstrating that the primary performance improvement of few-shot prompt stems from describing the space of questions and answers rather than direct Q&A mapping.

*Remark* 3 (Alignment of position). We claim semantic alignment is a more realistic setting for our T-C framework. However, employing semantic alignment introduces several challenges, such as describing position correspondence and considering the variations in position encoding. Nevertheless, we find it reasonable to embrace token-wise alignment, as we believe that a well-trained model can automatically bridge the gap between these two settings during training. By disregarding position offsets irrelevant to semantics, token-wise alignment can be achieved for semantically aligned samples through the introduction of blank characters.

From a training point of view, the model may be able to quickly learn to what extent the position offset is irrelevant to semantics, such as an additional space, so it does not affect any representation and generation. With this ability, if the model fits well on one training sample, it can also fit well on another sample, which is only different in some position offsets. Based on the observation, we can assume the existence of the latter sample (i.e., the sample with the position offsets) and use the token-wise alignment assumption.

As an additional explanation for this ability, we believe that the semantic and position information are disentangled in the Transformer. From a simple test, it is easy to test that the semantic embedding and position embedding are roughly orthogonal for most open-source models and therefore disentangled, which means the model can capture the semantic information and position information separately. Second, some results are also shown that the semantic information and the position information can be learned by different heads. See the analysis in Voita et al. (2019).

*Remark* 4 (Output probability and diversity). Another simplification in our theoretical framework pertains to the sampling scheme. We assume there is one *standard* token at each position while a more realistic setting involves the model's output being a probability distribution over the vocabulary and the output token is then sampled from the distribution. With random sampling, the model can generate diverse output with the same input.

Fortunately, our framework readily accommodates this sampling approach with simple modifications. We posit the existence of a distribution over the sequence space, which can be learned from a sufficiently large corpus. Given the prompt, question, and partial sequence, the distribution is projected into a conditional probability, describing the probability of the subsequent tokens. By replacing the individual training sample (or the sample set, if considering the hierarchical template) with this distribution, our framework seamlessly adapts to this setting. Formulating the sampling process into our framework will be our future work.

## D    EXPERIMENTS DETAILS AND MORE RESULTS

### D.1    CONCATENATE-LAST-LETTER DATASET

The template of the concatenate-last-letter dataset is:

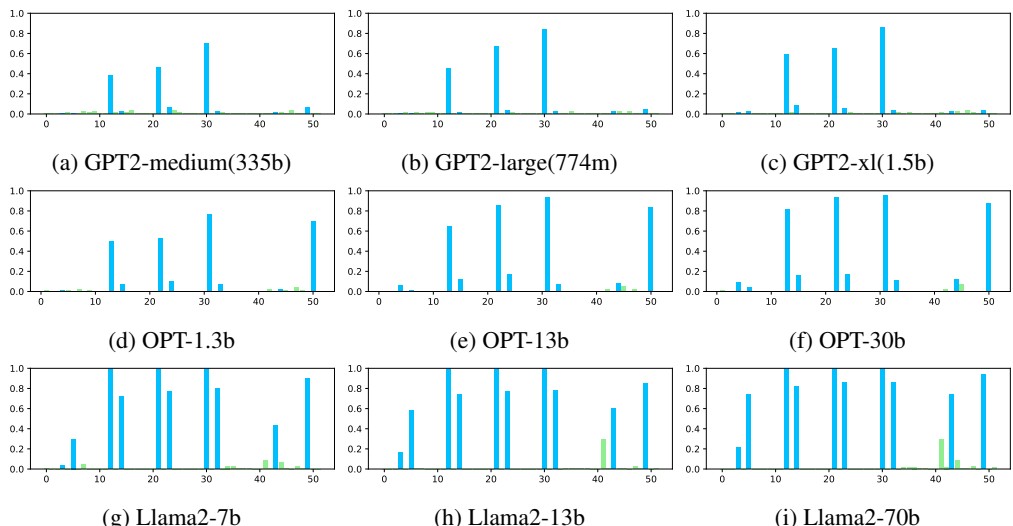

Figure 6: More results of the variance of the output distributions at template and content positions. X-axis: index of the tokens, y-axis: coefficient of variation. Blue bars: content tokens, a total of 10; green bars: template tokens.

Concatenate the last letters of the given words: `<word1>`, `<word2>`, `<word3>`, `<word4>`.
Let's think step by step.
1. The last letter of `<word1>` is `<letter1>`.
2. The last letter of `<word2>` is `<letter2>`.
3. The last letter of `<word3>` is `<letter3>`.
4. The last letter of `<word4>` is `<letter4>`.
5. Concatenating these letters together, we get `<answer>`.
Therefore, the answer is `<answer>`.

We produce the dataset by: (1) We collect the top 5,000 most commonly occurring English words from wiktionary[7]. (2) Randomly sample words as `<word>` and extract the corresponding letters and results.

### D.2 VARIANCE OF OUTPUT

More results of the coefficient of variance are shown in Figure 6. Here all of Llama-2 models we used are fine-tuned by chat data, i.e., the Llama-2-xxb-chat-hf model proposed by Huggingface. The CV of content (blue) is almost larger than template (green). The ROC curve is shown in Figure 7.

We also test on some other answer sequences generated by GPT-4 on the concatenate-last-letter task and follow the same process (labeling the content, replacing words and letters). These datasets differ in the specific template and content list but follows the same generating process. These templates are as follows:

Concatenate the last letters of the given words: `<word1>`, `<word2>`, `<word3>`, `<word4>`.
Let's think step by step.
1. Word: `<word1>`, last letter: `<letter1>`.
2. Word: `<word2>`, last letter: `<letter2>`.
3. Word: `<word3>`, last letter: `<letter3>`.
4. Word: `<word4>`, last letter: `<letter4>`.
Now, let us concatenate the last letters of each word: `<letter1>` + `<letter2>` + `<letter3>` + `<letter4>` = `<answer>`. Therefore, the concatenated result is `<answer>`.

The results are shown in Figure 8 and 9. The conclusion is the same as we shown in Figure 6 and in Section 6.1.

---

[7]https://en.wiktionary.org/wiki/Wiktionary:Frequency_lists/English/Wikipedia_(2016)

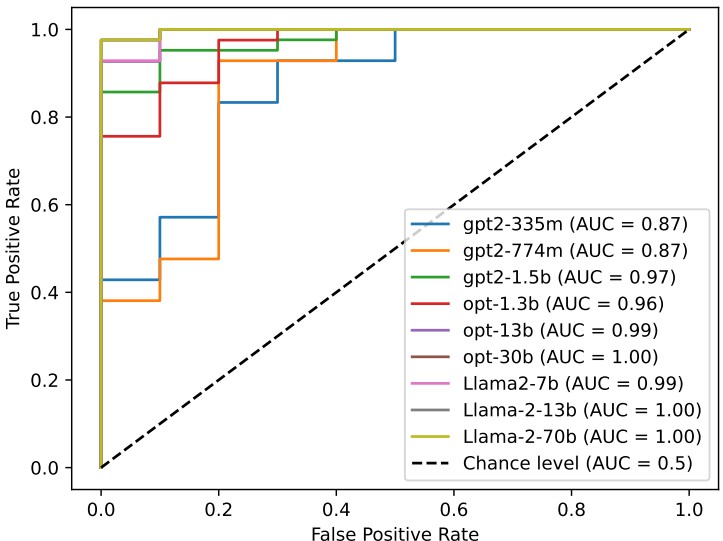

Figure 7: Receiver Operating Characteristic (ROC) of template tokens and content tokens.

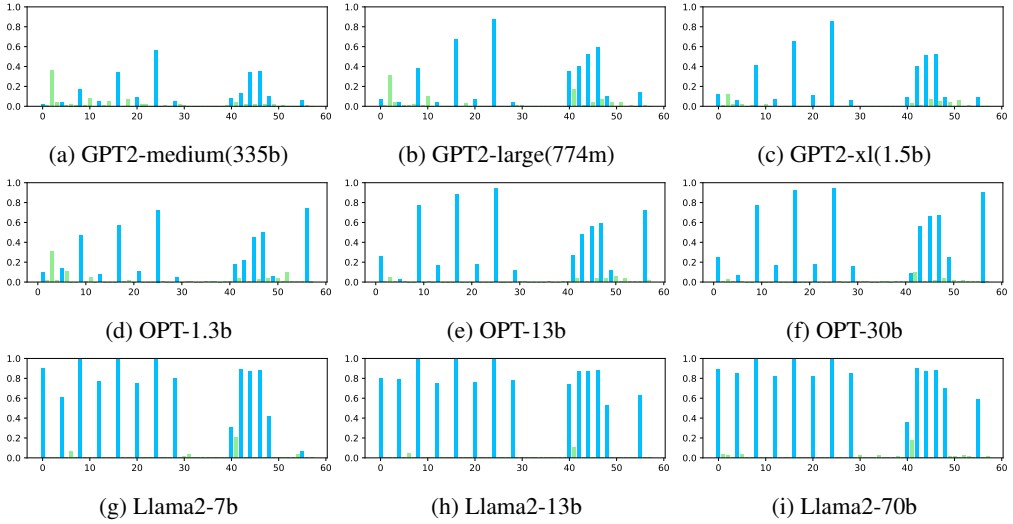

Figure 8: More results of the variance of the output distributions at template and content positions with **another template**. X-axis: index of the tokens, y-axis: coefficient of variation. Blue bars: content tokens, a total of 14; green bars: template tokens.

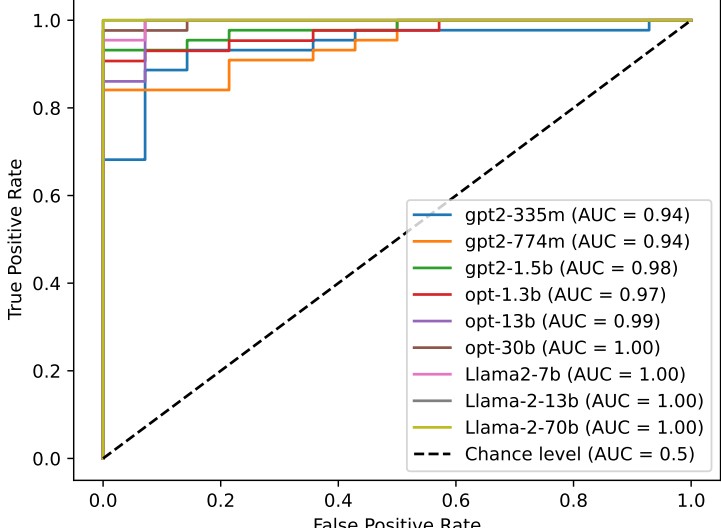

Figure 9: Receiver Operating Characteristic (ROC) of template tokens and content tokens with **another template**.

### D.3 VARIANCE-BASED T/C CLASSIFICATION

Here we exhibit some details of our variance-based T/C classifier.

#### D.3.1 WORD-LEVEL ANALYZER

As mentioned in Section 6.1, we need a word-level analyzer to address the issue of token-level alignment that cannot be performed on real datasets. With this analyzer, we define "words" and ensure that the tokenizers used by the practical models adhere to **the "sub-word" assumption**, meaning that there is no token spanning across two "words". In practice, we found that GPT-4 uses the coarsest-grained tokenizer. So, we first use GPT-4's tokenizer to divide sentences into different tokens. For each token, we check whether its first position is a whitespace or some punctuation, including periods, commas, colons, and semicolons. If it is, we keep it as a single token; otherwise, we concatenate it with the preceding token. This process gives us a word-level analyzer, guaranteeing that each token (generated by a tokenizer of a practical model such as Llama-2 or GPT-3) corresponds to a single "word" (defined by our word-level analyzer). When we say "word", we always mean a token split by our word-level analyzer.

#### D.3.2 ALGORITHM

The pseudo-code of the autoregressive T-C Classifier based on variance is shown in Algorithm 1.

#### D.3.3 FIRST-TOKEN-BASED CLASSIFICATION

We measure variance only on the first token for the T/C classification, and use it to represent the entire word. In most cases, determining the start of a word is sufficient to predict the entire word generation. However, in some cases, this may lead to the incorrect identification of $C$ as $T$. For example, when the generated content includes a pair of quotation marks and the tokenizer treats a single quotation mark as a separate token, the model recognizes the word as content (so the variance should have been higher) while the generation at the first position is fixed (a quotation mark) and the variance is low. It should be noted that simply splitting it into different words might violate the sub-word assumption mentioned above. Therefore, we choose to filter out some meaningless tokens from the output probabilities. Specifically, in the concatenate-last-letter dataset, we remove single whitespaces, line breaks \n, and a space-prefixed left quotation mark. In the SingleEQ dataset, we additionally remove space-prefixed dollar signs $ which are used to represent numbers in some training samples. For these tokens to be removed, we consider two methods. The first is to simply set

---

**Algorithm 1** Autoregressive T-C Classifier based on variance

---

**Require:** sentence $s$ with prompt $p$ as the beginning part, autoregressive model $M$, threshold $\theta$, replacing times $N$

**Ensure:** Classified sentence

1: Initialize empty lists: $T, C$
2: Initialize empty set for sequences with content replacement: $S \leftarrow ((), \ldots, ())$ ($N$ empty sequences.)
3: **for** each word $w$ in prompt $p$ **do**
4:     **Manually** determine whether $w$ belongs to $T$ or $C$.
5:     **for** $s' \in S$ **do**
6:         **if** $w$ belongs to $T$ **then**
7:             $s' \leftarrow s' + w$
8:         **else**
9:             **Manually** write the replaced content $replaceContent$
10:             $s' \leftarrow s' + replaceContent$
11:         **end if**
12:     **end for**
13: **end for**
14: $s \leftarrow s - p$
15: **while** $s$ is not empty **do**
16:     $w \leftarrow s[0], P \leftarrow \{\}$
17:     **for** each sentences $s'$ in $S$ **do**
18:         Record the output distribution: $P.\text{add}(\mathcal{M}(s'))$
19:     **end for**
20:     $variance \leftarrow$ measure the variance in the distribution set $P$.
21:     **if** $variance > \theta$ **then**
22:         Classify $w$ as content: $C.\text{add}(w)$
23:         **for** each sentences $s'$ in $S$ **do**
24:             $replaceContent \leftarrow \text{ArgMax}(\mathcal{M}(s'))$
25:             $s' \leftarrow s + replaceContent'$
26:         **end for**
27:     **else**
28:         Classify $w$ as template: $T.\text{add}(w)$
29:         Add $w$ to each sentences in $S$.
30:     **end if**
31:     $s \leftarrow s[1 :]$
32: **end while**
33: **return** $T, C$

---

[**Prompt**]: Concatenate the last letters of the given words: exception, currency, t-shirt, symbolic. Let's think step by step.
[**Answer**]:
1. Word: exception, last letter: n.
2. Word: currency, last letter: y.
3. Word: t-shirt, last letter: t.
4. Word: symbolic, last letter: c.
5. Now, let us concatenate the last letters of each word: n + y + t + c = nytc. together, we get the answer: nytc. Therefore, the concatenated result is ytge.

Figure 10: The classification results of the concatenate-last-letter dataset. Threshold: 0.4. Template: yellow, content: blue. We mark the token whose classification conflicts with the human intuition as red.

their value to a very small number and then re-normalize through the softmax function. The second is to continue considering the generation at the next position: we first set the probability (after softmax) of these filtered tokens as zero. Then we just set the current position's output as one of the tokens to be removed and get the generation distribution at the next position. We multiply this output's probability distribution by the probability of this token in the original output and then add it to the original distribution. For example, if we want to remove whitespace's probability from the current probability, denoted as $P$, we first set its probability to zero, denoted it as $\hat{P}$. And then we assume the current output is just the whitespace and make the model generate the output probability at the next position, denoted as $P'$. Then we multiply this distribution by the probability of the whitespace $P(\text{whitespace}) \cdot P'$ and add it to $\hat{P}$. It is easy to check the sum of the new distribution is still one. This procession is just like skipping the whitespace and replacing the next token in the current place. The latter method is more accurate in practice, preventing the filtered tokens from dominating the probabilities, but it significantly reduces inference speed. We only use the latter method in "SingleEQ" and we also set a threshold ($1\%$) to pass this procession when the probability of the filtered token does not exceed a certain value.

### D.3.4 CONTENT GENERATION

Another issue is content generation. Since we no longer use human annotations and instead rely on the model's judgment of when to generate content, we use the model itself to generate tokens that should be filled in under different content replacements. The challenge here is how to determine whether content generation has finished. For example, in the concatenate-last-letters dataset, when the model detects the next token after "3. The last letter of" should be content, the model should generate some tokens that will be used to fill in this position while detecting the subsequent positions. However, the generated `<word3>` could be tokenized into several tokens and cannot be generated in one step. So we need a criterion to detect whether the generation has finished. Here, we assume one content consists of a **single** word. We use the same criterion as our word-level analyzer mentioned above, which is to keep adding generated tokens until a token with a space (or punctuation, line break, etc.) is generated. At this point, all previously generated tokens can be combined into a "word" and used as a replacement for the current content.

### D.3.5 MORE RESULTS

More results of the T/C classification are shown in Figure 10, 11 and 12.

[**Prompt**]: Concatenate the last letters of the given words: electric, plain, apartment, brain. Let's think step by step.
[**Answer**]:
1. Extract the last letters of each word: electric: c, plain: n, apartment: t, brain: n.
2. Concatenate the extracted letters: c + n + t + n = cntn. together, we get the answer: cntn. Therefore, the concatenated result is cntn.

Figure 11: The classification results of the concatenate-last-letter dataset. Threshold: 0.3. Template: yellow, content: blue. We mark the token whose classification conflicts with the human intuition as red.

[**Prompt**]: Solve the arithmetic problem. Question: There were 6 roses in the vase. Mary cut some more roses from her flower garden. There are now 16 roses in the vase. How many roses did she cut ? Let's think step by step.
[**Answer**]:
1. Start with the number of roses in the vase before Mary cut any: 6.
2. We wonder how many roses Mary cut from her garden. Let's call this number "x".
3. Add the number of roses Mary cut to the original number in the vase: 6 + x.
4. This equals the total number of roses in the vase after Mary cut some: 16.
5. So we can set up an equation: 6 + x = 16.
6. To solve for x, we need to isolate it on one side of the equation. We can do this by subtracting 6 from both sides: x = 16 - 6 = 10.
7. Therefore, Mary cut 10 roses from her garden. Therefore, the answer (arabic number) is 10.

Figure 12: The classification results of the SingleEQ dataset. Threshold: 0.35. Template: yellow, content: blue. We mark the token whose classification conflicts with the human intuition as red.

