# OpenReview forum: "Explaining the Complex Task Reasoning of Large Language Models with Template-Content Structure"
_ICLR.cc/2024/Conference — Submitted to ICLR 2024_

### Official Review · Reviewer_KnhL · 2023-10-31

**Soundness:** 3 good
**Presentation:** 3 good
**Contribution:** 3 good
**Rating:** 6
**Confidence:** 3

**Summary:**

The authors present a structured framework in this paper to elucidate the workings of language models, emphasizing a template-task structure for answer generation. The paper is articulate and well-structured. I do have some inquiries concerning its content

**Strengths:**

The explanations in the paper are clearly written in simple terms.

**Weaknesses:**

The authors frequently relegate crucial details to the appendix, disrupting the continuity of the reading experience.

Related papers:

Explaining Agent Behavior with Large Language Models - Link: https://arxiv.org/pdf/2309.10346.pdf

iPrompt: Explaining Data Patterns in Natural Language via Interpretable Autoprompting - Link: https://arxiv.org/pdf/2210.01848.pdf

It would be valuable to examine cases with misclassification errors to assess the robustness of the proposed model.

**Questions:**

Are there alternative explanation models in the existing literature? A detailed comparison, possibly supplemented with numerical experiments, would be beneficial.

---

> ### Author Response · Authors · 2023-11-21
>
> Thank you for your thoughtful and comprehensive review. Your feedback provides us with clear direction on how to improve our article. We summarize your concerns and suggestions as well as our responses as follows：
>
> 1. > "The authors frequently relegate crucial details to the appendix, disrupting the continuity of the reading experience."
>
>     Thank you for your valuable feedback. In our endeavor to present a completely novel perspective on LLMS, we encountered the challenge of incorporating motivation, examples, mathematical deduction, and experimental evidence within the confines of a succinct 9-page document. This limitation unavoidably constrained our capacity to delve into comprehensive details in the main body of the text. To facilitate readers in quickly comprehending our t-c structure, we gave precedence to elucidating the motivation and intuition. As a result, more intricate details were placed in the appendix, catering to readers with specific interests.
>
>     In the final version of our paper, we intend to refine the organization further. We are particularly grateful for any feedback, and we highly value your insights regarding the elements that should receive greater emphasis in the main text. Your suggestions are instrumental in enhancing the quality of our work.
>
>
> 2. **Related work**
>     > "Related papers:
>     >
>     > Explaining Agent Behavior with Large Language Models - Link: https://arxiv.org/pdf/2309.10346.pdf
>     >
>     > iPrompt: Explaining Data Patterns in Natural Language via Interpretable Autoprompting - Link: https://arxiv.org/pdf/2210.01848.pdf"
>
>     Thank you for your kind reminder. These papers indeed offer a valuable addition to our current section on related work. Contrary to the works we have cited that focus on elucidating LLMs themselves, these studies explore a distinct facet of LLM interpretability: employing LLMs to explain external systems. We will incorporate these references into the related work section of our final version.
>
>     - The first paper "*Explaining Agent Behavior with Large Language Models*" introduces a methodology for generating model-agnostic explanations for agent behaviors using LLMs. This approach involves translating an agent's decision-making process into a decision tree, which is then inputted into an LLM to produce explanations. Their crowd experiments show that LLMS generate better explanations than previous work.
>     - The second work "*iPrompt*" propose to find the pattern in the dataset, i.e. utilizing LLMs to elucidate the dataset itself. This method involves feeding "input-output" pairs ${(x_1,y_1),\dots,(x_n,y_n)}$ into an LLM, resulting in the generation of semantic explanations for the relationships between $x$ and $y$ in natural language. The explanations thus generated have been highly rated in terms of human evaluative metrics.
>
>     Both works show that LLMs can summarize and explain the input, including dataset patterns and the output of external systems. It suggests that LLMs learn a "concept" or "task" that is more abstract than specific data, which is evidence for our point.

---

> > ### Author Response · Authors · 2023-11-21
> >
> > 3. > "It would be valuable to examine cases with misclassification errors to assess the robustness of the proposed model."
> >
> >     Thank you for highlighting this crucial aspect. We have indeed conducted a deeper examination to understand instances where the t/c classification of LMs diverges from human intuition, which leads to more interesting results. Our observations include:
> >     - First, the classification results are fairly conclusive for most positions, which means the variance at each positions is either close to 0 (i.e. a definitive template token) or a fairly large value (i.e. a content token), which means for most positions the t/c classification is clear.
> >     - Ambiguous or misclassified results tend to occur in scenarios where two semantically identical language usages result in different classifications. For instance, consider the sentence "*John gave 43 seashells to Sam.*" (in Figure 5, right), where LMs give different classification on tokens 'John', 'to' and 'Sam':
> >       - Recall that our classification model cannot see the current word but only the previous words to classify the current position to be generated. For the position "*John*", the LM is not sure whether a vocabulary like "*Therefore*" should be generated here. There is a mixture of two kinds of generation --- "John" or "Therefore, John". So it gives a relatively small variance (a value between obvious content and template).
> >       - Similarly, the phrases "*gave 43 seashells to Sam.*" and "*gave Sam 43 seashells.\n*" are semantically equivalent, but correspond to "TCCTCT" and "TCCCTT" respectively. It leads to the misclassification of the words "*to*" and "*Sam*", which are the fourth and fifth tokens.
> >     - Such "misclassification" actually contains more inspiration than just "error", such as the "*habit*" of an LM. For example, the above example shows that the LM actually prefers to add more adverbs or conjunctions at the beginning of sentences to clearly indicate the logical structure.
> >     - This classification model can be improved in the future. A three-class model can distinct these ambiguous positions and label them as "ambiguous" instead of "template" or "content" to further check them manually or using other models.

---

> > > ### Author Response · Authors · 2023-11-21
> > >
> > > 4. > "Are there alternative explanation models in the existing literature? A detailed comparison, possibly supplemented with numerical experiments, would be beneficial."
> > >
> > >     We recognize and appreciate the importance of contrasting alternative explanation models with our template-content (t-c) structure. The comparison can be outlined as follows:
> > >
> > >     1. Some work studies the theoretical expressive power of LLMs from a complexity theory perspecitve. These studies reveal the complexity classes — such as Turing Completeness [1], P-Completeness [2], or TC-Completeness [3] — that LLMs are capable of expressing, given specific conditions like window lengths, variable precision, and depth. While these insights are valuable in understanding the potential of LLMs, they primarily focus on theoretical models tailored for particular abstract operations, without direct application to practical models and tasks. In contrast, our work is anchored in language models pre-trained on natural language tasks and CoT-like step-by-step reasoning, thus offering a more practical perspective.
> > >     2. Some research employs case studies to demonstrate that LMs can perform specific tasks like "linear regression" [4] or "gradient descent" [5]. These studies often necessitate meticulous design to implement certain mathematical operations, diverging from practical scenarios. Our work, not relying on the simulation of any specific operation in concrete tasks, is more general and closer to the practical setting.
> > >     3. The study [6] explains the in-context CoT ability from the Bayesian viewpoint, positing that LMs acquire a "concept" attribute by learning long-range dependencies during training. We can liken our 'template' to a textual representation of this "concept." However, this approach falls short in explaining how models generate correct answers as token sequences or in zero-shot scenarios.
> > >     4. There is also some work [7] trying to visualize the attention map or activation value of LMs to explain their behavior. But these works often focus on one specific model (for example, a pretrained GPT2-medium) and some naive tasks and need help to provide a general understanding. The visualization becomes more complicated when the model becomes larger and the task becomes more complex.
> > >
> > >     Given the diversity in focus areas, phenomena, models, and tasks of these explanation models, it is challenging to definitively compare them or design a numerical experiment for a fair comparison, as they do not share identical concerns. However, we emphasize that our work is the first to analyze the behavior of LMs from a template-content perspective to give it a formal definition and theoretical analysis. Our experiments have given preliminary evidence that our interpretation is consistent with the performance of actual LMs. A new way of understanding contributes to the development of LMs.
> > >
> > > **Reference**:
> > >
> > > [1] Perez, Jorge, Pablo Barcelo, and Javier Marinkovic: Attention Is Turing Complete, 2021. <https://jmlr.org/papers/v22/20-302.html>.
> > >
> > > [2] William Merrill, Ashish Sabharwal: The Expresssive Power of Transformers with Chain of Thought, 2023. <https://arxiv.org/abs/2310.07923>.
> > >
> > > [3] Luo Shengjie, Shanda Li, Shuxin Zheng, Tie-Yan Liu, Liwei Wang, and Di He: Your Transformer May Not Be as Powerful as You Expect, 2022. <https://openreview.net/forum?id=NQFFNdsOGD>.
> > >
> > > [4] Ekin Akyürek, Dale Schuurmans, Jacob Andreas, Tengyu Ma, Denny Zhou: What Learning Algorithm is In-Context Learning? Investigations with Linear Models, ICLR 2023. <https://arxiv.org/abs/2211.15661>.
> > >
> > > [5] Kwangjun Ahn, Xiang Cheng, Hadi Daneshmand, Suvrit Sra: Transformers learn to implement preconditioned gradient descent for in-context learning, NeurIPS 2023. <https://arxiv.org/abs/2306.00297>.
> > >
> > > [6] Sang Michael Xie, Aditi Raghunathan, Percy Liang, Tengyu Ma: An Explanation of In-context Learning as Implicit Bayesian Inference, ICLR 2022. <https://arxiv.org/abs/2111.02080>.
> > >
> > > [7] Michael Hanna, Ollie Liu, Alexandre Variengien: How does GPT-2 compute greater-than?: Interpreting mathematical abilities in a pre-trained language model, NeurIPS 2023. <https://arxiv.org/abs/2305.00586>

---

> ### Author Response · Authors · 2023-11-30
> **Summary of our response**
>
> Dear Reviewer KnhL,
>
> We trust this message finds you in good health. The author response period for our submission has been extended until the end of December 1st, due to the paper not having received three reviews yet.
>
> As we are nearing the end of this extension period, we note with concern that we have not yet received your feedback. In preparation for the rebuttal, we have exerted substantial effort to enhance our submission, specifically focusing on:
>
> 1. Engaging in a thorough discussion of some pertinent related work.
> 2. Conducting a detailed analysis of the misclassification error in the t-c classification.
> 3. Making a comprehensive comparison between our t-c structure and other explanations in the field.
>
> We would greatly appreciate it if you could inform us whether our revisions and responses have adequately addressed your concerns, or if you have any further questions or points for clarification.
>
> Your time and dedication in reviewing our work are greatly valued and appreciated.
>
> Sincerely,
>
> Authors

---

### Official Review · Reviewer_KmxN · 2023-11-04

**Soundness:** 3 good
**Presentation:** 2 fair
**Contribution:** 2 fair
**Rating:** 5
**Confidence:** 2

**Summary:**

This paper aims to show the existence of the template-content decomposition of complex natural language tasks. The paper first provides proof for the UAT of casual transformers, and then proves the existence of template transformers and content transformers. The paper is not easily readable. Even though the existence of the transformers model is proved, it is not clear the implication of the theorem. This is because the Transformers function $f_T$ and $f_C$ could be much more complex than a single transformer model. Also, there is no experimental result showing such decomposition benefits the tasks either in computation efforts or task performances.

**Strengths:**

* Provide proof of the existence of template transformers and content transformer functions for the template-content decomposition

**Weaknesses:**

* The importance of the existence of template-content decomposition is unknown
* There is no experimental result showing such decomposition benefits the tasks either in computation efforts or task performances.

**Questions:**

* Can $f_T$ and $f_C$ be more complex than a single transformer model? In other words, to decompose tasks that can be handled by an LLAMA 7B model, will the $f_T$ and $f_C$ be an LLAMA 70b function?

* Will such model decomposition benefit the tasks either in computation efforts or task performances? Also, will such decomposition work for more general NLP tasks such as summarization, translation, or LLM benchmark tasks including [HELM](https://crfm.stanford.edu/helm/latest/) or [lm-evaluation-harness](https://github.com/EleutherAI/lm-evaluation-harness)

---

> ### Author Response · Authors · 2023-11-23
>
> Thank you for your insightful comments. We address your concerns as follows:
>
> 1. **The importance of the existence of template-content decomposition**:
>     > "It is not clear the implication of the theorem (the existence of template transformers and content transformers). The importance of the existence of template-content decomposition is unknown."
>
>     The t-c structure leads to **the reasoning ability** of modern LLMs, and we prove it in proposition 1 (section 4.4). The skeleton of our paper is to prove that (1) the t-c structure leads to reasoning ability, i.e., generating answers for complex reasoning problems, and (2) the transformers can learn the t-c structure. Combining these two parts, we can explain the reasoning ability of LLMs. Below is a detailed explanation.
>
>     Intuitively, one key point of complex reasoning lies in learning the correct combination of the multiple small steps, i.e., **the skeleton of a solving process**. This observation can be easily verified by examining answers generated by humans or LLMs. Taking math problems as an example, the process of human-generated answers typically consists of two parts: the overall problem-solving skeleton and the specific solution of each step. At the same time, powerful LLMs like chatGPT also show similar behavior (as shown in Figure 1). The skeleton determines the type of the problem and the solution method (such as a linear equation system composed of two variables in our example in Figure 1) while the specific steps perform concrete math operations (for example, determine the sum of two variables x + y is **35**). In our t-c structure, the skeleton tokens are defined as **template** and the specific step tokens are defined as **content**. The template should stay the same once the problem type is determined, while the content can vary according to concrete problems.
>
>     In Figure 1, we show such an important example to help understand the importance of the t-c structure. In the classic [rabbit and chicken problem](http://mathelogical.in/non-routine-mathematics/the-famous-rabbit-and-chicken-problem/), even if we change rabbit and chicken to correct and wrong answers (and also change the number of feet to the number of marks), the LLM uses the same template (of linear equation system) to solve the problem. In other words, the template remains the same for all problems of the same type, while the content is different for each problem. Therefore, we emphasize that if an LLM can generalize on various problems of the same type, rather than just remembering individual data points that appear in the training set, it **must be able to learn the template (invariant part) of the task**. Otherwise, if we modify the problem slightly, it will fail. So, the existence of the t-c structure is the key to the reasoning ability of LLMs.
>
>     Our theoretical analysis (sections 4.3 and 4.4) is to express the above intuition formally. Section 4.3 proves the existence of such a t-c transformer. Then, Proposition 1 (in section 4.4) shows the t-c transformers can generate the answer for a complex reasoning problem as long as they have seen a similar problem of the same task (even though its content may be completely different from the target problem), which paves the way for LLMs' generalizable reasoning ability.
>
>     In summary, template-content decomposition is important because it explains where LLMs' strong reasoning ability comes from and shows that the transformer architecture can learn such a t-c structure. We illustrate it intuitively and theoretically in our paper. And we further verify it with additional experimental evidence in our response to the third point.
>
> 2. **The complexity of the template (or content) transformers**:
>     > "The template (and content) Transformers function could be much more complex than a single transformer model... Can $f_T$ and $f_C$ be more complex than a single transformer model? In other words, to decompose tasks that can be handled by a LLAMA 7B model, will the $f_T$ and $f_C$ be an LLAMA 70b function?"
>
>     No. Suppose a model can indeed handle the task, which means that any changes to the content of the problem will not affect its solution process. In this case, this model has successfully learned the t-c structure because our t-c structure is precisely defined by the independence of the template from variations in content. That is, if a Llama 7B model handles a task, it must have achieved the decomposition within its 7B parameters---a small model that can solve a type of problem indicates that this type of problem is not complex.

---

> ### Author Response · Authors · 2023-11-23
>
> 3. **Experimental results to show the importance of t-c decomposition**
>     > There is no experimental result showing such decomposition benefits the tasks either in computation efforts or task performances.
>
>     Thank you for your kind reminder. We are glad to explain our experimental results further and add some experimental evidence to show the importance of t-c decomposition.
>
>     1. First, our experimental results in section 6 show **the correlation between the capacity of LM to handle t-c structure with their reasoning performance**, which means for current LMs, a significant t-c structure is a condition of complex reasoning ability. To show the relation more clearly, we choose the t-c classification **AUC-ROC** as the t-c classification accuracy. We also report the **difference of the mean-variance** (**DMV**) $\sum_{s\in C} V_{s} / N_C-\sum_{s\in T} V_{s} / N_T$ where $N_T$ ($N_C$) is the number of template (content) tokens, and $V_s$ is the variance of the output probability at position $s$. DMV can measure how confident the model is about their classification results because the larger the DMV is, the more significant the different behaviors of the model for T and C. We compare them to their complex reasoning (CoT) ability and accuracy on GSM8k as the measure of their complex reasoning ability.
>
>        | Model | DMV | AUC-ROC | CoT ability | Accuracy on GSM8k*** |
>        | ----------- | ----------- | ----- | ----- | --- |
>         | GPT2-335m | 0.15 | 0.87 | $\times$ | 0.0 |
>         | GPT2-774m | 0.28 | 0.87 | $\times$ | 0.0 |
>         | GPT2-1.5b | 0.27 | 0.97 | $\times$ | 0.0 |
>         | OPT-1.3b | 0.29 | 0.96 | $\times$* | 0.0 |
>         | OPT-13b | 0.42 | 0.99 | $\times$* | 0.0 |
>         | OPT-30b | 0.46 | 1.00 | $\times$* | 0.0 |
>         | Llama2-7b | 0.72 | 0.99 | $\checkmark$** | 14.6 |
>         | Llama2-13b | 0.80 | 1.00 | $\checkmark$** | 28.7 |
>         | Llama2-70b | 0.81 | 1.00 | $\checkmark$** | 56.8 |
>
>         \* The OPT models have trouble following the instructions or simulating the examples of prompts; therefore, no CoT ability is shown. See [1] as a reference.
>
>         \*\* Llama-2 models have CoT ability to do some complex reasoning, such as GSM8k or MATH. See [2] for reference.
>
>         \*\*\* Test in the 8-shot CoT setting to align with our framework. The CoT accuracy of the GPT2 and OPT models without CoT ability is 0.
>
>         The results show that the CoT ability, i.e., the complex reasoning ability, is available only for models that handle significant t-c structures. It shows that the t-c structure is the key to LLMs' complex reasoning (or CoT ability). For the model with less t-c classification ability, such as GPT2s, generating a reasonable CoT is too hard because just one wrong token will lead to failure under autoregressive generation, so the accuracy is 0.
>
>     2. The t-c structure can help the reasoning performance of LMs. For example, the template-content structure can be used as **data augmentation**, where we can replace the content in the training data to instruct our LMs to learn the decomposition and better reasoning ability.
>     Here, we collect 100 examples of the '*chicken and rabbit problem*' to form a dataset. For each question, we collect five answers generated by GPT-4. We use these question-answer pairs to finetune a Llama2-7b model. As a comparison, we train another Llama2-7b model with the same dataset and other settings but randomly **replacing the content** (denoted as **cr**) in each question-answer pair as data augmentation. In order to ensure that our content replacement does not just mitigate overfitting due to the introduction of data perturbation, we also conducted experiments on models with general data augmentation. Here, we selected EDA[3] as the baseline, a data augmentation method based on synonym replacement, random word deletion, and addition.
>     The efficacy of these approaches is depicted in our [Figure](https://imgur.com/a/R2cl6Ef), where the training steps are plotted along the x-axis, and the test accuracy (reported as pass@3) is represented on the y-axis. The results clearly demonstrate that content replacement enhances model performance. Specifically, the maximum performance increased from 75% (vanilla model, blue curve) to 85% (cr, orange curve), and from 85% (eda, green curve) to 95% (eda+cr, red curve) in the models incorporating cr augmentation. This experiment demonstrates that teaching the model the T-C structure through content replacement indeed helps the model to reason better, showcasing the importance of the T-C structure in reasoning.

---

> ### Author Response · Authors · 2023-11-23
>
> 4. **Other task**
>
>    > "Will such decomposition work for more general NLP tasks such as summarization, translation, or LLM benchmark tasks including HELM or lm-evaluation-harness?"
>
>     Yes. While primarily demonstrated on complex reasoning tasks like arithmetic and symbolic reasoning, our template-content decomposition approach can apply to more conventional NLP tasks, provided these tasks incorporate relatively standardized processes. Take, for example, the task of summarizing a lengthy article. This task can be approached by initially extracting key information from each paragraph and then synthesizing these elements into a coherent summary. A possible template for this process might be: "The first paragraph discusses [...]. The second paragraph addresses [...]," and so forth. This method can be further granularized to summarizing individual sentences within a paragraph, using sub-templates like: "Based on the sentence \<sentence\>, it appears to focus on \<subject\>."
>
>     Another example could be commonsense reasoning such as [OpenbookQA](https://paperswithcode.com/dataset/openbookqa)[4] dataset. Here we show an answer generated by GPT-4.
>
>     >Question:
>     >Which of these would let the most heat travel through?
>     >A) a new pair of jeans.
>     >B) a steel spoon in a cafeteria.
>     >C) a cotton candy at a store.
>     >D) a calvin klein cotton hat.
>     >
>     >ChatGPT:
>     >
>     >To determine which of these items would allow the most heat to travel through, we can analyze each option based on the thermal conductivity of the materials involved:
>     >
>     >A) A new pair of jeans: Jeans are typically made from denim, a material woven from cotton fibers. Cotton is a poor conductor of heat, making it a good insulator. This means that jeans would not allow much heat to travel through them.
>     >
>     >B) A steel spoon in a cafeteria: Steel is a metal and metals are generally good conductors of heat. This means that a steel spoon would allow heat to travel through it quite efficiently.
>     >
>     >C) A cotton candy at a store: Cotton candy is made from sugar that is heated and spun into thin strands. While sugar can absorb some heat, the airy structure of cotton candy makes it a poor conductor of heat. It would not allow much heat to travel through it.
>     >
>     >D) A Calvin Klein cotton hat: Similar to the jeans, a hat made of cotton would also be a poor conductor of heat, as cotton is a good insulating material.
>     >
>     >Based on this analysis, option B) a steel spoon in a cafeteria, would let the most heat travel through due to the high thermal conductivity of steel.
>
>     This answer contains four steps that analyze each of the choices and a result. Each analyzing step can be decomposed into: 1) The index, 2) the object, 3) the property of this object, 4) the relation between the property and the question. LMs can use the same process to answer similar questions.
>
>     For these tasks, our t-c structure can also be applied. However, for some tasks like translation, the process is not fixed and the template is unclear, which is beyond our coverage.
>
> [1] Susan Zhang, Stephen Roller, et al: OPT: Open Pre-trained Transformer Language Models, 2022. <https://arxiv.org/pdf/2205.01068.pdf>.
>
> [2] Hugo Touvron, Louis Martin, et al: Llama 2: Open Foundation and Fine-Tuned Chat Models, 2023. <https://arxiv.org/pdf/2307.09288.pdf>.
>
> [3] Jason Wei, Kai Zou: EDA: Easy Data Augmentation Techniques for Boosting Performance on Text Classification Tasks, 2019. <https://arxiv.org/abs/1901.11196>.
>
> [4] Todor Mihaylov, Peter Clark, Tushar Khot, Ashish Sabharwal: Can a Suit of Armor Conduct Electricity? A New Dataset for Open Book Question Answering, EMNLP 2018. <https://arxiv.org/abs/1809.02789>.

---

> ### Author Response · Authors · 2023-11-30
> **Summary of our response**
>
> Dear Reviewer KmxN,
>
> We trust this message finds you in good health. The author response period for our submission has been extended until the end of December 1st, as it has not yet received three reviews.
>
> As we approach the end of this extension period, we note with concern that your feedback has not been received. In preparation for the rebuttal, we have dedicated considerable effort to address key aspects of our work, including:
>
> 1. Elucidating the importance of the t-c decomposition, that the decomposition leads to the reasoning ability.
> 2. Providing a explanation of the complexity in each component of our model.
> 3. Presenting additional experimental results that reinforce the validate of the t-c structure.
> 4. Offering a explanation of the specific tasks for which our methodology is most appropriate.
>
> We would be immensely grateful if you could inform us whether our responses have satisfactorily addressed your concerns, or if there are any further queries you might have.
>
> Your engagement and thorough evaluation of our work are highly valued and appreciated.
>
> Kind regards,
>
> Authors

---

### Official Review · Reviewer_oieR · 2023-11-19

**Soundness:** 2 fair
**Presentation:** 2 fair
**Contribution:** 2 fair
**Rating:** 3
**Confidence:** 4

**Summary:**

This paper proposes a framework to explain language model's ability to solve complex reasoning tasks as parsing the input and generating the output in a "template-content" structured way. The paper theoretically formalizes this framework, and extends the framework to the hierarchical version to explain why models can solve arbitrarily complex tasks. Finally, the authors used the last-letter-concatenation task to show their T-C framework can help explain models' reasoning abilities in practice.

**Strengths:**

- The proposed template-content framework is interesting, and the extension to the hierarchical version is also more flexible in explaining model's abilities in solving complex reasoning tasks.

**Weaknesses:**

This paper made many simplified (possibly unrealistic) assumptions in its proposed theoretical framework and misses the connection to real-world tasks.
- For example, the paper assumes a model is "well trained", meaning the model can memorize all answers perfectly. The authors claimed that this assumption is "not challenging for the prevailing LLMs with a huge amount of parameters", but many existing work has shown that even the largest models could struggle with tail knowledge [1].
- The hierarchical extension assumes a model can decompose complex tasks well into sub-template and sub-content. But as the sub-template and sub-content can have many different combinations (especially if they are in slightly different forms as defined in Appendix A.1, based on label consistency), it is unrealistic to assume that the model is able to search in this combinatorial space efficiently and find the right template/content combination, especially when the task is very complex. Thus the presented theory in its current form, doesn't seem to be sufficient in explaining model's ability to solve very complex tasks.

[1] Kandpal et al. Large Language Models Struggle to Learn Long-Tail Knowledge. ICML 2023.

In addition, the experiments are rather weak and do not support the theoretical part very well.
- The two tasks used are very simple tasks, last-letter-concat and SingleEQ. In order to support the authors' claim on the *flexibility* and *generalizability* of this framework, a slightly more complex task should be presented as well. For example, on slightly more difficult algorithmic reasoning tasks like GSM8K or AQuA, can the hierarchical T-C be used to explain model's reasoning?
- The results on T/C classification is a bit ad-hoc. Based on Figure 4 right, 1) as the authors mentioned, some green bars are too short to see, so it is unclear which model "exhibits the clearest T/C distinction"; 2) can the authors provide more quantitative analysis on the ratio between the green/blue bars? from the 2nd figure and the 3rd figure, the ratios seem not very significantly different, so I'm not sure if one can indeed judge a model has a better "reasoning capability" from those ratios.
- In Figure 5, even on the simple SingleEQ task, the proposed classification already conflicts with the human intuition, so I'm not sure if this can be used in practice, especially for more complex reasoning tasks. Also on this task, the authors define content as "names, objections, and Arabic numbers", this also seems very ad-hoc and how can one apply this in general for any tasks?


Missing references, the following paper also discusses how text (similar to the "content" part defined in this paper) and patterns (similar to the "template" part defined in this paper) affect reasoning performance in language models:
- Text and Patterns: For Effective Chain of Thought, It Takes Two to Tango. Madaan et al. 2022.

**Questions:**

- Can the authors provide more quantitative analysis on the ratio between the green/blue bars in Figure 4?
- How can one define template/content in general for any tasks?

---

> ### Author Response · Authors · 2023-11-25
>
> Thank you for your insightful comments and suggestions. We appreciate the opportunity to elucidate and clarify our article further. Below, we provide a summary of your questions and suggestions:
>
> 1. **Reasonability of the assumptions**:
>    > "This paper made many simplified (possibly unrealistic) assumptions in its proposed theoretical framework and misses the connection to real-world tasks. For example, the paper assumes a model is "well trained", meaning the model can memorize all answers perfectly. The authors claimed that this assumption is "not challenging for the prevailing LLMs with a huge amount of parameters", but many existing work has shown that even the largest models could struggle with tail knowledge [1]."
>
>     As you rightly pointed out, it is crucial for a theoretical analysis to pay attention to whether the assumptions are reasonable, and we pay close attention to this. We do not assume that the model can memorize all answers perfectly. We only need the LM to memorize just **one** answer, which can provide the template for the target task. For example, to answer an arithmetic problem, the model learns the template (or sub-template in our hierarchical setting) from **anyone** correct solution in its training set. Therefore, our assumption is much weaker and more practical than "learning tail knowledge".
>     Also, our t-c structure is proposed to *explain* whether and how LMs can handle generalized reasoning. Here, "generalized" denotes the ability of models to not only recall but also to process problems of a similar nature with varying content. Suppose a model cannot memory any problem of a certain task in the training set. In that case, it is almost impossible to reason on this task, so our explanation model is naturally not suitable for such a model --- *"explanation" is valid only for those who do have the ability to reason*.
>
>    > "The hierarchical extension assumes a model can decompose complex tasks well into sub-template and sub-content. But as the sub-template and sub-content can have many different combinations (especially if they are in slightly different forms as defined in Appendix A.1, based on label consistency), it is unrealistic to assume that the model is able to search in this combinatorial space efficiently and find the right template/content combination, especially when the task is very complex. Thus the presented theory in its current form, doesn't seem to be sufficient in explaining model's ability to solve very complex tasks."
>
>     No, our t-c framework **does not involve search**. Our theory does not require models to search or find a combination, but emphasizes that *natural language itself has the property to recombine fragments from different texts to form new sequences*. Because of **the composability of natural language**, models can generate the correct combined answer to solve the task, as long as these examples which can be combined into an answer **exist**, as our Proposition 2 says.
>
>     Here, we explain the composability more intuitively. Please consider this arithmetic problem in GSM8k:
>     > Randy has 60 mango trees on his farm. He also has 5 less than half as many coconut trees as mango trees. How many trees does Randy have in all on his farm?
>
>     To solve the problem, LMs need first solve the half of mango tress numbers: 60/2=30, which can be learned from a sample that contains the template about half: "*half ... /2=*" (which translates the word 'half' into an equation '*.../2*') and another one contains "*60/2=30*" (solving the equation). As long as the intermediate result *30* has been generated, the template '*\<num a\> less than \<num b\>... \<num b\> - \<num a\>*'' with the content *5* and *30* in this problem leads to the equation *'30-5'* where the answer *25* is learned from any sample contains the equation, no matter what this sample is originally about. The following steps are similar.
>     From this process, we find that the answer to this problem can be split into different template fragments, each of which can come from different training samples. It is this property (separability or composability) that ensures that our model can naturally generate this answer by learning these samples separately. For the model itself, we have no requirements other than remembering these samples.

---

> ### Author Response · Authors · 2023-11-25
>
> 2. **Experiment**:
>    > "The two tasks used are very simple tasks, last-letter-concat and SingleEQ. In order to support the authors' claim on the flexibility and generalizability of this framework, a slightly more complex task should be presented as well. For example, on slightly more difficult algorithmic reasoning tasks like GSM8K or AQuA, can the hierarchical T-C be used to explain model's reasoning?"
>
>     We provide an intuitive example above to show that the hierarchical t-c structure can be used to explain the reasoning ability on GSM8k. However, it is challenging to design an experiment to show the practical models' behavior on such a complex task, because there could be too many levels and too many possible reasonable decompositions which prevent us from making a clear intuitive judgment.
>     In fact, mostly focusing on providing a formal and theoretical explanation, our experiments aim to support our framework, at least on the most typical t/c tokens. We acknowledge our experiments are preliminary. As we say in the limitation section, We expect this theoretical explanation to be mutually reinforcing with newer visualization work.
>
>    > "The results on T/C classification is a bit ad-hoc. Based on Figure 4 right, 1) as the authors mentioned, some green bars are too short to see, so it is unclear which model "exhibits the clearest T/C distinction"
>    > "2) Can the authors provide more quantitative analysis on the ratio between the green/blue bars? from the 2nd figure and the 3rd figure, the ratios seem not very significantly different, so I'm not sure if one can indeed judge a model has a better "reasoning capability" from those ratios .... Can the authors provide more quantitative analysis on the ratio between the green/blue bars in Figure 4?"
>
>    Yes, we explain the results in Figure 4 right further here. Firstly, we should compare them by the *difference* instead of the *ratio* because these variances are based on (normalized) probabilities, i.e., the values after softmax. For any model, the variance at each position has the same range: $[0, (N-1)/N]$ where the $N$ is the number of disturbances. (Due to our y-axis range, it looks like some of the bars exceed 1 incorrectly. Actually, they are just very close to 1. We will modify this in the final version to make it more intuitive.) We no longer need to eliminate the impact of scale by calculate the ratio. On the contrary, since the variance of the template is often close to 0, the ratio would excessively amplify the differences between insignificant small quantities. For example, there is actually not much difference between $10^{-3}$ and $10^{-4}$.
>
>    Here, we focus on whether a threshold can separate them. So we give an AUC-ROC score (Appendix D.2) and the **difference of the mean-variance** (**DMV**) $\sum_{s\in C} V_{s} / N_C-\sum_{s\in T} V_{s} / N_T$ where $N_T$ ($N_C$) is the number of template (content) tokens, and $V_s$ is the variance of the output probability at position $s$. AUC-ROC score measures the t-c classification accuracy, and the DMV further measures how confident models are about their classification. We also report the test accuracy on GSM8k as measures of their reasoning ability.
>
>    | Model | DMV | AUC-ROC | CoT ability | Accuracy on GSM8k*** |
>    | ----------- | ----------- | ----- | ----- | --- |
>     | GPT2-335m | 0.15 | 0.87 | $\times$ | 0.0 |
>     | GPT2-774m | 0.28 | 0.87 | $\times$ | 0.0 |
>     | GPT2-1.5b | 0.27 | 0.97 | $\times$ | 0.0 |
>     | OPT-1.3b | 0.29 | 0.96 | $\times$* | 0.0 |
>     | OPT-13b | 0.42 | 0.99 | $\times$* | 0.0 |
>     | OPT-30b | 0.46 | 1.00 | $\times$* | 0.0 |
>     | Llama2-7b | 0.72 | 0.99 | $\checkmark$** | 14.6 |
>     | Llama2-13b | 0.80 | 1.00 | $\checkmark$** | 28.7 |
>     | Llama2-70b | 0.81 | 1.00 | $\checkmark$** | 56.8 |
>
>     \* The OPT models have trouble following the instructions or simulating the examples of prompts; therefore, no CoT ability is shown. See [1] as a reference.
>
>     \*\* Llama-2 models have CoT ability to do some complex reasoning, such as GSM8k or MATH. See [2] for reference.
>
>     \*\*\* Test in the 8-shot CoT setting to align with our framework. The CoT accuracy of the GPT2 and OPT models without CoT ability is 0.
>
>     These results show the evident tendency that the significant t-c structure is the prerequisite for strong reasoning ability.

---

> ### Author Response · Authors · 2023-11-25
>
> 2. **Experiment**: (contd.)
> > "In Figure 5, even on the simple SingleEQ task, the proposed classification already conflicts with the human intuition, so I'm not sure if this can be used in practice, especially for more complex reasoning tasks."
>     In our shown example in SingleEQ, only three "misclassifications" exist among the 77 tokens.
>     Ambiguous or misclassified results tend to occur in scenarios where two semantically identical language usages result in different classifications. For instance, consider the sentence "*John gave 43 seashells to Sam.*" (in Figure 5, right), where LMs give different classification on tokens 'John', 'to' and 'Sam':
>       - Recall that our classification model cannot see the current word but only the previous words to classify the current position to be generated. For the position "*John*", the LM is not sure whether a vocabulary like "*Therefore*" should be generated here. There is a mixture of two kinds of generation --- "John" or "Therefore, John". So it gives a relatively small variance (a value between obvious content and template).
>       - Similarly, the phrases "*gave 43 seashells to Sam.*" and "*gave Sam 43 seashells.\n*" are semantically equivalent, but correspond to "TCCTCT" and "TCCCTT" respectively. It leads to the misclassification of the words "*to*" and "*Sam*", which are the fourth and fifth tokens.
>     - Such "misclassification" actually contains more inspiration than just "error", such as the "*habit*" of an LM. For example, the above example shows that the LM actually prefers to add more adverbs or conjunctions at the beginning of sentences to clearly indicate the logical structure.
>
>
>     We acknowledge this classification model can be improved. A three-class model can distinct these ambiguous positions and label them as "ambiguous" instead of "template" or "content" to check them further manually or using other models. However, because we use this classification model as a tool to show the reasonability of our t-c structure in this paper, we leave it as future work.
>
>     > "Also on this task, the authors define content as "names, objections, and Arabic numbers", this also seems very ad-hoc and how can one apply this in general for any tasks?"
>
>     Thank you for your reminder. We do not expect our framework can apply for any task. Instead, we focus only on reasoning tasks that have relatively obvious fixed processes. Here, we choose "names, objections and Arabic numbers" as the typical content in some arithmetic problems. These contents are most commonly replaced to generate similar problems. For example, these two questions in GSM8k are the same except for different names, verbs (but the same mathematical meaning), objections, and Arabic numbers.
>
>     > Natalia sold clips to 48 of her friends in April, and then she sold half as many clips in May. How many clips did Natalia sell altogether in April and May?
>     >
>     > Salvadore earned $1956. Santo earned half of what Salvadore earned. How much money did the two men earn together?
>
>     And these problems could be further extended to more complex problems in GSM8k:
>
>     > Hash has nine more than half as many toys as Bill has. If Bill has 60 toys, how many total toys do the boys have?
>     >
>     > Caleb has 3 dozen jellybeans. Sophie has half as many jellybeans as Caleb. How many jellybeans do they have in total?
>     >
>     > Randy has 60 mango trees on his farm. He also has 5 less than half as many coconut trees as mango trees. How many trees does Randy have in all on his farm?
>     >
>     > Lizzie has half as many crayons as Bobbie. Bobbie has three times as many crayons as Billie. If Billie has 18 crayons, how many crayons does Lizzie have?
>
>     In summary, we acknowledge that these are only part of all the possible content, but they are still representative for some tasks.

---

> ### Author Response · Authors · 2023-11-25
>
> 3. > "How can one (define template/content) and apply this (t-c classification model) in general for any tasks?"
>
>    We are not aiming to define the t-c structure for any tasks. Instead, we aim to explain the complex reasoning ability of LLMs, i.e., the reasoning involving a relative fixed logical structures and steps. Most arithmetic reasoning (such as singleEQ, GSM8k, ...), most symbolic reasoning (last-letter-concatenation), some commonsense reasoning, and some classic NLP tasks (such as summarization) can be explained by our t-c structure.
>
>    Because we have shown some examples of arithmetic and symbolic reasoning in our paper. Here we give more examples of other tasks. Take summarization as an example. This task can be approached by initially extracting key information from each paragraph and then synthesizing these elements into a coherent summary. A possible template for this process might be: "The first paragraph discusses [...]. The second paragraph addresses [...]," and so forth. This method can be further granularized to summarizing individual sentences within a paragraph, using sub-templates like: "Based on the sentence \<sentence\>, it appears to focus on \<subject\>."
>
>     Commonsense reasoning such as [OpenbookQA](https://paperswithcode.com/dataset/openbookqa)[3] dataset could be also with a template. Here we show an answer generated by GPT-4.
>
>     >Question:
>     >Which of these would let the most heat travel through?
>     >A) a new pair of jeans.
>     >B) a steel spoon in a cafeteria.
>     >C) a cotton candy at a store.
>     >D) a calvin klein cotton hat.
>     >
>     >ChatGPT:
>     >
>     >To determine which of these items would allow the most heat to travel through, we can analyze each option based on the thermal conductivity of the materials involved:
>     >
>     >A) A new pair of jeans: Jeans are typically made from denim, a material woven from cotton fibers. Cotton is a poor conductor of heat, making it a good insulator. This means that jeans would not allow much heat to travel through them.
>     >
>     >B) A steel spoon in a cafeteria: Steel is a metal and metals are generally good conductors of heat. This means that a steel spoon would allow heat to travel through it quite efficiently.
>     >
>     >C) A cotton candy at a store: Cotton candy is made from sugar that is heated and spun into thin strands. While sugar can absorb some heat, the airy structure of cotton candy makes it a poor conductor of heat. It would not allow much heat to travel through it.
>     >
>     >D) A Calvin Klein cotton hat: Similar to the jeans, a hat made of cotton would also be a poor conductor of heat, as cotton is a good insulating material.
>     >
>     >Based on this analysis, option B) a steel spoon in a cafeteria, would let the most heat travel through due to the high thermal conductivity of steel.
>
>     This answer contains four steps that analyze each of the choices and a result. Each analyzing step can be decomposed into: 1) The index, 2) the object, 3) the property of this object, 4) the relation between the property and the question. LMs can use the same process to answer similar questions.
>
>    However, for some tasks like translation, the process is not so fixed, and the template is unclear, which is beyond our coverage.

---

> ### Author Response · Authors · 2023-11-25
>
> 4. > "Missing references, the following paper also discusses how text (similar to the "content" part defined in this paper) and patterns (similar to the "template" part defined in this paper) affect reasoning performance in language models: Text and Patterns: For Effective Chain of Thought, It Takes Two to Tango. Madaan et al. 2022."
>
>     Thank you for your kind reminder. It is a fascinating work focusing on explaining the CoT ability of LLMs, which is indeed related to our work. We have some similar inspirations. For example, their symbol can be seen as part of our content, and they say the symbol replacement will not degrade the performance, which is consistent with our Proposition 1. We are glad to add it to our related work. Unsurprisingly, some work have similar motivations to us because it's so natural. But these two papers are still different enough, as follows:
>
>     1. With a similar idea to decompose different roles in the CoT, their pattern, symbol and text cannot directly correspond to our template and content. For example, in their mathematical examples, the equation "5+4=9" is a pattern, which is similar to our "content" because the number and the operator depend on the concrete problem. However, in their date example, their pattern consists of implicit steps: \<restate\>, \<intermediate results\> and \<output\>, which corresponds to our template.
>     2. They only give some example tasks and do not formally define the symbol, pattern, and text in general tasks. In our work, one main contribution is providing a formal definition and theoretical analyzing tool.
>     3. As other work from in-context learning to explain the reasoning ability, their work focuses on how the few-shot exemplars motivate models to reason, while our work is not limited to few-shot. The examples in the paper demonstrate that our t-c structure can explain the reasoning ability under zero-shot (instruction).
>     4. Because of different decomposition, their conclusion differs from ours. They focus on the mutual help between pattern and text, but we emphasize the uni-direction dependence between templates and content (or templates at different levels).
>
> [1] Susan Zhang, Stephen Roller, et al: OPT: Open Pre-trained Transformer Language Models, 2022. <https://arxiv.org/pdf/2205.01068.pdf>.
>
> [2] Hugo Touvron, Louis Martin, et al: Llama 2: Open Foundation and Fine-Tuned Chat Models, 2023. <https://arxiv.org/pdf/2307.09288.pdf>.
>
> [3] Todor Mihaylov, Peter Clark, Tushar Khot, Ashish Sabharwal: Can a Suit of Armor Conduct Electricity? A New Dataset for Open Book Question Answering, EMNLP 2018. <https://arxiv.org/abs/1809.02789>.

---

> ### Author Response · Authors · 2023-11-30
> **Summary of our response**
>
> Dear Reviewer oieR,
>
> We trust this communication finds you in good health. The author response deadline for this submission has been extended until December 1st end-of-day, due to the paper not receiving three reviews.
>
> As the extended deadline approaches, we have yet to receive your valuable feedback. We have invested great effort in our rebuttal, which includes:
>
> 1. Addressing and clarifying two critical misunderstandings regarding our assumptions, along with a explanation about their rationality.
> 2. Providing additional experimental data and analyses to elucidate the t-c classification outcomes.
> 3. Elucidating the tasks for which our approach is most suited.
> 4. Engaging in a discourse on a recent piece of related work.
>
> We would greatly appreciate it if you could inform us whether these responses have addressed your concerns, or if there are any further queries you wish to raise.
>
> Your time and effort in reviewing our work are deeply appreciated.
>
> Kind regards,
> The Authors

---

> > ### Comment · Reviewer_oieR · 2023-11-30
> >
> > Thanks for the response. However, I think the authors still need to provide a more rigorous study on the t-c structure and provide stronger evidence through experiments in order to support their claims.
> >
> > 1) Memorization of templates and search:
> > - for the mango tree example provided, how do you ensure the model has seen the template that "half" = "X/2"? (It's probably true for large enough models trained on a wide variety of texts, but what if a model is slightly smaller, or trained on books without seeing any math problems?) In addition, what if instead of "half", the question asks about 1/3 or 1/4, or even 1/11? How do you know a "well-trained" model can memorize those templates or not?
> > - For search, it is possible that the same problem can be decomposed into different templates, with some templates are well memorized by the model while some are not, how can you ensure the model can find the right template combination that they have memorized well? Again using the mango tree example, why not use "say X is the number of coconut trees, then 2(X+5)=60" if the model is trained on a lot of algebra data?
> > - I think in order to support the authors' claim, the authors should provide stronger evidence on what has / hasn't been memorized by each model, and how the model efficiently searches and combines the well-memorized templates to solve a problem (or the inability to do so).
> >
> >
> > 2) flexibility and generalizability of the t-c structure:
> > - Thanks for adding the summarization example. However, a lot of real-world summarization corpus do not necessarily use this kind of structure, e.g., CNN/DailyMail, or XSum. I think the authors need to provide more extensive analysis on how their t-c structure supports a larger set of tasks, instead of using very simple tasks and very simple structures, which puts the flexibility and generalizability of the proposal into question.

---

> > > ### Author Response · Authors · 2023-12-01
> > >
> > > Thank you for your response! We am very pleased to have the opportunity to further explain our work. We hope these responses can further address your questions, and we welcome your continued inquiries and suggestions.
> > >
> > > 1. > for the mango tree example provided, how do you ensure the model has seen the template that "half" = "X/2"? (It's probably true for large enough models trained on a wide variety of texts, but what if a model is slightly smaller, or trained on books without seeing any math problems?) In addition, what if instead of "half", the question asks about 1/3 or 1/4, or even 1/11?
> > >
> > >     Previous work has shown that they have less reasoning ability because these models are smaller or trained on books without seeing any math problems. However, our t-c structure aims to explain the reasoning ability of models, which indeed have the corresponding ability. The situation for "1/3", "1/4" or "1/11" is similar. If fewer occurrences exist in the training set, LMs cannot solve the problem. In this situation, we cannot explain how the LMs solve the problem.
> > >
> > >     To verify this idea, we operate a toy text where we input the sentence "Assuming the number of bananas is x, the half of the bananas is " to a Llama-2-13b model. Among 10 tests, the LM can always (10/10) generate the correct answer "x/2" or "0.5x"; for "the quarter", the accuracy is 2/10, and for "one eleventh", the accuracy is 0/10. When we search in the GSM8k training set, the word "half" occurs 1551 times, while "quarter" occurs 303 times and "eleventh" occurs just one time. The observation verifies that having seen and learned the corresponding sample is the **condition** to finish a task. For models that are not large enough to learn the sample or have not seen the corresponding samples, these LMs do not have the reasoning ability for these tasks and are unsuitable for our t-c structure.
> > > 2. > How do you know a "well-trained" model can memorize those templates or not?
> > >     - We do not define a "wel-trained" model but a model is well-trained on a sample, where a model is well-trained **on a sample** just when the model can memorize this sample. As we claimed in the first point, memorization is the condition to have the reasoning ability for a model. So we can assume for these models with reasoning ability on a task, i.e., the models we focus on, the assumption that the model has memorized at least one sample is reasonable.
> > >     - Because most of the current LMs are closed-source for their training data, we cannot easily check whether one training sample is memorized. However, as shown above, the correlation between the frequency of occurrence and the accuracy of generating the corresponding template is evident. This supports our claim that if a model cannot even memorize a training sample, it cannot generalize this task. Therefore, the "well-trained" assumption is reasonable for these models that we focus on and have corresponding reasoning abilities.
> > >     - On the other hand, what we care about is the generalization ability of the model on different problems within the task, and generalization ability is more difficult than memory ability. It is generally believed that generalization should be based on memory. In various works [1,2,3] studying generalization ability, it is common to assume that memory on the training set has been achieved. The related work [4] uses the same assumption that models have fit the pretrain distribution perfectly to explain the in-context learning ability.

---

> > > ### Author Response · Authors · 2023-12-01
> > >
> > > 3. > For search, it is possible that the same problem can be decomposed into different templates, with some templates are well memorized by the model while some are not, how can you ensure the model can find the right template combination that they have memorized well? Again using the mango tree example, why not use "say X is the number of coconut trees, then 2(X+5)=60" if the model is trained on a lot of algebra data?
> > >
> > >     Yes, there are many possible decompositions for a task, and any one of them can lead to an answer. A model with algebra ability can use "2(x+5)=60" to solve the task. In fact, our theory does not require a unique decomposition of a task.
> > >
> > >     In one combination, some levels cannot find a memorized sample. In this case, the model is indeed unable to reason correctly. It is unsurprising because even the most powerful LMs today cannot achieve 100% accuracy on complex inference tasks. However, what combinations a model tends to generate depends on what top-level template is generated because the template decides the steps (i.e., combinations). And what the template the model selects depends on how the model was trained. The phenomenon we observed is that for models with stronger reasoning performance (such as GPT-4), they always tend to split the tasks very carefully to avoid subtasks that are too complex to find suitable training samples to solve - this may be related to high-quality training samples or RLHF. For these models, they memorize a large number of templates and generate more detailed and reasonable splits, which ensures their generation accuracy.
> > > 4. > The authors should provide stronger evidence on what has / hasn't been memorized by each model, and how the model efficiently searches and combines the well-memorized templates to solve a problem (or the inability to do so).
> > >
> > >     Examining the internal mechanisms of LMs is a very challenging task, and to date, we have yet to see any satisfactory work that provides visualizations about what facts LMs remember or what features they extract during reasoning. Therefore, it is difficult to conduct direct experiments to verify this. Still, the evidence presented above suggests that assuming their memory capability is reasonable as a premise for generalization. For the combination, as long as LMs learn a good template, the combination should be reasonable, and each level template is easy to learn. The combination is natural in all of the examples we have provided, and each part of the sub-task is easy to learn.
> > > 5. > However, a lot of real-world summarization corpus do not necessarily use this kind of structure, e.g., CNN/DailyMail, or XSum.
> > >
> > >     Yes, if the task does not involve step-by-step reasoning or the step-by-step reasoning cannot help the task, our t-c structure is unsuitable. Our t-c structure focuses on those tasks with a fixed process to solve. Our example points out that not only arithmetic problems involve step-by-step reasoning and thus can be suitable for our model. It doesn't stress that every task can benefit from this.
> > > 6. >  The authors need to provide more extensive analysis on how their t-c structure supports a larger set of tasks, instead of using very simple tasks and very simple structures, which puts the flexibility and generalizability of the proposal into question.
> > >
> > >     Our T-C structure suits complex tasks requiring task decomposition and step-by-step reasoning. These tasks should have a more apparent structural nature, meaning that different problems within the same task have the same problem-solving steps. Such tasks are not all-encompassing, but it is satisfied in arithmetic reasoning, symbolic reasoning, and some natural language tasks. As for flexibility and generalizability, what we mean is that templates at different levels do not necessarily need to come from the same sample but can be combined across different samples. About this idea, we have provided examples in GSM8k, a more complex arithmetic dataset, which is far away from being solved by current LMs. Our restrictions on templates will not be too excessive for some tasks involving multi-level reasoning (such as mathematical problems requiring multiple steps of calculation). The flexibility and generalizability mentioned here are separate from spanning across multiple tasks.
> > >
> > > [1] Zeyuan Allen-Zhu, Yuanzhi Li, Yingyu Liang: Learning and Generalization in Overparameterized Neural Networks, Going Beyond Two Layers, NeurIPS 2018.
> > >
> > > [2] Behnam Neyshabur, Zhiyuan Li, Srinadh Bhojanapalli, Yann LeCun, Nathan Srebro: Towards Understanding the Role of Over-Parametrization in Generalization of Neural Networks, 2018.
> > >
> > > [3] Trevor Hastie , Robert Tibshirani , Jerome Friedman: The Elements of Statistical Learning Data Mining, Inference, and Prediction, Second Edition. 2009
> > >
> > > [4] Xie, Sang Michael, Aditi Raghunathan, Percy Liang, and Tengyu Ma: An Explanation of In-context Learning as Implicit Bayesian Inference, 2022.

---

### Author Response · Authors · 2023-11-25
**The discussion phase has been extended! Welcome to your reply!**

Dear Reviewers,

We are immensely grateful for your insightful comments and suggestions on our paper. We are pleased to have been informed that **our discussion phase has been extended to December 1st**.

We deeply value your opinions and are looking forward to further discussions with you. Should you have any more questions or suggestions, we eagerly await your feedback. Your contributions are highly appreciated, and we thank you once again for your dedication and expertise.

Best regards,

Authors

---

### Author Response · Authors · 2023-11-28
**Looking Forward to Further Discussion!**

Dear reviewers,

We are truly grateful for the time and effort you've invested in reviewing our work. Your valuable insights and feedback have greatly enhanced our understanding and perspective on our research. With your comments and suggestions in mind, we have prepared a comprehensive rebuttal addressing each point you raised.

As the discussion phase concludes in three days, we hope for a chance to engage in further dialogue. We believe that some aspects of the feedback could greatly benefit from additional discussion, ensuring a thorough understanding for all parties involved.

Once again, we extend our heartfelt thanks for your consideration and for the prospect of further discussing our work with you.

Best regards,

Authors
Authours

---

### Meta-Review · Area_Chair_PGGG · 2023-12-07

**Metareview:**

This paper proposes a framework that models answer generation for complex natural language tasks as a hierarchical "template-content" structure and investigates the complex reasoning abilities of large language models. However, after a careful reading of the reviews, I would no recommend acceptance of the paper due to several concerns, including the unclear importance of the template-content decomposition, lack of experimental results demonstrating its benefits in computational efforts or task performances, and many simplified or unrealistic assumptions in the proposed theoretical framework. The paper's current form does not seem to provide a sufficient explanation for a model's ability to solve complex tasks, as it assumes perfect memorization of answers and efficient search in the combinatorial space of sub-template and sub-content combinations.

**Justification For Why Not Higher Score:**

n/a

**Justification For Why Not Lower Score:**

n/a

---

### Decision · Program_Chairs · 2024-01-16

Reject